# A comparative genomics methodology reveals a widespread family of membrane-disrupting T6SS effectors

Chaya M. Fridman[1], Kinga Keppel[1], Motti Gerlic [1], Eran Bosis [2✉] & Dor Salomon [1✉]

Gram-negative bacteria deliver effectors via the type VI secretion system (T6SS) to out-compete their rivals. Each bacterial strain carries a different arsenal of effectors; the identities of many remain unknown. Here, we present an approach to identify T6SS effectors encoded in bacterial genomes of interest, without prior knowledge of the effectors' domain content or genetic neighborhood. Our pipeline comprises a comparative genomics analysis followed by screening using a surrogate T6SS+ strain. Using this approach, we identify an antibacterial effector belonging to the T6SS1 of *Vibrio parahaemolyticus*, representing a widespread family of T6SS effectors sharing a C-terminal domain that we name Tme (Type VI membrane-disrupting effector). Tme effectors function in the periplasm where they intoxicate bacteria by disrupting membrane integrity. We believe our approach can be scaled up to identify additional T6SS effectors in various bacterial genera.

[1] Department of Clinical Microbiology and Immunology, Sackler Faculty of Medicine, Tel Aviv University, Tel Aviv 6997801, Israel. [2] Department of Biotechnology Engineering, ORT Braude College of Engineering, Karmiel 2161002, Israel. ✉email: bosis@braude.ac.il; dorsalomon@mail.tau.ac.il

The type VI secretion system (T6SS) is a protein secretion apparatus that is widespread in Gram-negative bacteria[1–3]. It can deliver toxins, called effectors, directly into bacterial and eukaryotic neighbors; thus, it can play a role in both bacterial competition and virulence[4–7]. T6SS effectors decorate a tail tube structure that is propelled from the bacterial cell upon contraction of an engulfing sheath[8]. The tail tube comprises three structural components named Hcp, VgrG, and PAAR repeat-containing protein[2,3,8–10]. A VgrG trimer is sharpened by a PAAR protein to form a spike; the spike caps a tube of stacked hexameric rings formed by Hcp. These tail tube components can themselves function as effectors when they contain additional C-terminal toxin domains[4,10]. Other 'cargo effectors' can bind to one of these three tail components, either directly or with the aid of an adapter protein[11–15]. Effectors that mediate antibacterial activities are encoded adjacent to a cognate immunity protein that antagonizes self- and kin-intoxication[6,16].

The T6SS effector arsenal is dynamic. Isolates of the same bacterial species carry different T6SS effector repertoires[17–21]. This diversity has been associated with the horizontal transfer of effector genes, and it may also have arisen from effector gene duplications[19,22]. To date, several experimental and computational approaches have been utilized to identify T6SS effectors in a given bacterial strain. Experimental approaches, such as comparative proteomics[5,16,22–24] and transposon screens[25], were employed on several bacterial species; they revealed many effectors with both antibacterial and anti-eukaryotic activities. However, these methods are often laborious and time consuming, allowing the characterization of only a single strain at a time. Furthermore, these experimental approaches require that the strain in question be available and genetically tractable to generate T6SS mutants. In addition, they necessitate prior knowledge of the conditions that activate their T6SS. They may also fail to identify effectors that are expressed and delivered in low amounts under the examined experimental conditions. Computational approaches, including analyses of genes found inside T6SS clusters and auxiliary modules[14,20,26–29], genes encoding proteins with T6SS marker domain (e.g., MIX and FIX)[17,23,26], or genes that neighbor T6SS adapters (e.g., DUF4123, DUF1795, and DUF2169)[13,15,19,30–32], have resulted in the discovery of many effector families that have proven useful in preliminary analyses of new bacterial genomes[33]. However, such bioinformatic approaches may fail to identify "orphan effectors" that do not neighbor any known T6SS component on the genome, and that do not contain any known T6SS-associated domain. Therefore, additional methods are required to identify T6SS effectors in multiple bacteria that are not necessarily available or amenable to genetic manipulations, or for which the conditions required to activate T6SS remain unknown.

In this work, we seek to establish comparative genomics as a methodology that allows quick, unbiased identification of candidate T6SS effectors in multiple bacterial genomes. We rely on the premise that effectors of a given T6SS are genetically linked to the system itself, and therefore should not be found in genomes lacking that T6SS (Fig. 1a). Here, we focus on identifying antibacterial T6SS effectors, since they present a large pool of novel mechanisms and targets that can be used to develop new antibacterial treatments against multi-drug-resistant pathogens.

To demonstrate the feasibility of a comparative genomics approach in identifying T6SS antibacterial effectors, we use the Gram-negative, marine pathogen Vibrio parahaemolyticus[34], for which many high-quality genomic sequences are publically available. V. parahaemolyticus is a major cause of acute hepatopancreatic necrosis disease (AHPND) in shrimp[35,36], and of seafood-borne gastroenteritis[37,38]. Most isolates carry one or two T6SSs; T6SS1 is encoded by a gene cluster predominantly found in pathogenic isolates, whereas T6SS2 is found in all V. parahaemolyticus isolates[39,40]. T6SS1 is active under warm marine-like conditions, and it mediates antibacterial activity[26,40]. Three T6SS1 antibacterial effectors were identified in the clinical isolate RIMD 2210633: VP1388 and VP1415, which are encoded at both ends of the T6SS1 cluster (Supplementary Fig. 1), and VPA1263, which is encoded in a genomic island[23]. However, much less is known about T6SS1 and its effector repertoire in other V. parahaemolyticus strains.

To identify T6SS1 effectors in V. parahaemolyticus, we devise a methodology comprising two steps: a comparative genomics analysis to identify candidate effectors in a given bacterial genome, followed by a functional screen in a surrogate T6SS platform. Utilizing this pipeline to analyze the genome of the V. parahaemolyticus isolate BB22OP, we identify a family of T6SS effectors, which we name Tme (Type VI membrane-disrupting effector, as explained in the Results section); Tme members are widespread in many Proteobacterial genomes. We demonstrate that Tme effectors, which are genetically and functionally linked to T6SSs, function in the bacterial periplasm and disrupt membrane integrity. We also describe their cognate immunity family, named Tmi. Our findings confirm that this pipeline is a useful methodology that can enable large-scale identification of T6SS effectors.

## Results

**Identifying T6SS1 antibacterial effector candidates.** To establish a comparative genomics pipeline for identifying T6SS1 effectors in V. parahaemolyticus, we first compiled a dataset of high-quality V. parahaemolyticus genomes. This dataset comprised the publically available genomes of 175 V. parahaemolyticus isolates collected from around the world (Supplementary Data 1). Using the structural and regulatory genes of the previously characterized T6SS1 gene cluster in V. parahaemolyticus isolate RIMD 2210633[40] (vp1391-vp1414; Supplementary Fig. 1) as templates, we divided the genome dataset into T6SS1[+] (121 genomes) and T6SS1[−] (52 genomes) groups (Fig. 1b and Supplementary Data 2). Two genomes contained only a partial T6SS1 cluster, and were therefore excluded from subsequent analyses.

To test the comparative genomics approach, we analyzed the assembled genome of the V. parahaemolyticus isolate BB22OP[41], in which the T6SSs have not yet been studied. BB22OP harbors a T6SS1 gene cluster (VPBB_RS06640-VPBB_RS06785) highly similar to the previously studied RIMD 2210633 antibacterial T6SS1 cluster (Supplementary Fig. 1), including effectors that are encoded at the ends of the cluster[23,40]. However, we did not find known effectors outside the BB22OP T6SS1 cluster. Therefore, we hypothesized that the BB22OP genome harbors other effectors that diversify its T6SS1 effector repertoire.

We posited that candidate T6SS1 effectors are proteins for which closely related homologs are found only in T6SS1[+] genomes, and not in T6SS1[−] genomes. To identify such candidates, the amino acid sequences of 4724 BB22OP proteins were searched against our V. parahaemolyticus genome dataset for homologs. For each BB22OP protein, the similarity percentage value obtained from the closest homolog in each genome in the dataset was recorded (Fig. 1b and Supplementary Data 3). Candidate BB22OP T6SS1 antibacterial effectors were defined as proteins for which closely related homologs were only identified in T6SS1[+] genomes, and that maintained additional characteristics shared among experimentally validated T6SS antibacterial effectors: (i) candidates should not contain a canonical secretion signal, and (ii) candidate antibacterial effectors should be encoded adjacent to a gene transcribed in the same direction (putative

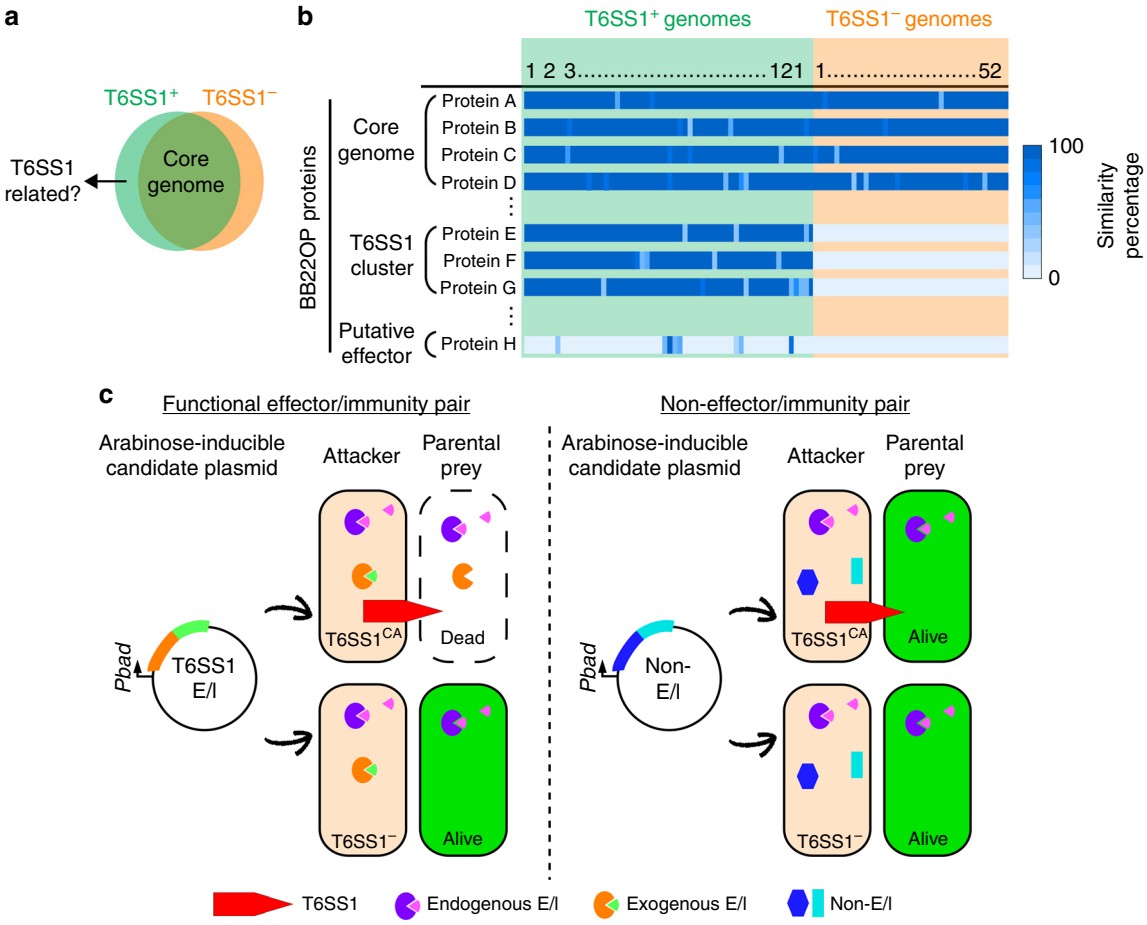

**Fig. 1 T6SS1 effector identification methodology. a** Venn diagram displaying the premise for using comparative genomics to identify T6SS1-related proteins and candidate effectors. **b** The first step in the T6SS1 effector identification methodology. An overview of the expected results from analyzing *V. parahaemolyticus* BB22OP proteins against the *V. parahaemolyticus* genome dataset; proteins belonging to the *V. parahaemolyticus* core genome are expected to have highly similar homologs in both T6SS1+ and T6SS1− genomes, whereas the closely related homologs of T6SS1 components and putative effectors should only be found in T6SS1+ genomes. **c** The second step in the T6SS1 effector identification methodology. Schematic representation of the *V. parahaemolyticus* surrogate T6SS1 platform. Candidate antibacterial T6SS1 effectors are cloned, together with a putative neighboring immunity, into an expression vector under *Pbad* regulation. Expression vectors are introduced into a strain with a constitutively active T6SS1 (T6SS1CA; the surrogate platform), and into a derivative mutant in which the T6SS1 is inactive (T6SS1−). The ability of these attacking strains to kill a competing parental prey strain in a T6SS1-dependent manner is monitored. The parental prey strain contains the same endogenous effector/immunity (E/I) pairs as the attackers. Thus, it can antagonize their attack if they did not acquire a genuine E/I pair; however, if the expression plasmid in the attacker encodes a genuine E/I pair, then the parental prey will not be able to resist T6SS1-mediated intoxication.

immunity). Notably, we did not consider proteins encoded within the T6SS1 gene cluster (VPBB_RS06640-VPBB_RS06785) as candidate novel effectors in this analysis, since they were homologs of T6SS1 components and effectors previously characterized in the RIMD 2210633 isolate. Of the 4724 proteins encoded on the BB22OP genome, we identified 17 that conformed to the above set of restrictions (Supplementary Data 4). Importantly, none of the genes encoding these proteins were found in proximity to T6SS-related genes (Supplementary Data 3). After manual assessment of these proteins and their genetic neighborhoods, we concluded that 10 proteins are probably not antibacterial effectors (see details in Supplementary Data 4). Thus, seven antibacterial T6SS1 effector candidates remained.

**Establishing a surrogate T6SS1 platform.** Next, we sought to screen the seven remaining candidates for antibacterial effectors that neighbor a cognate immunity gene. To circumvent the need for laborious and time-consuming construction of BB22OP

deletion strains that will enable self-competition experiments to characterize T6SS effector/immunity (E/I) pairs, we sought to establish a simple and fast screening assay to test candidate T6SS1 E/I pairs from any *V. parahaemolyticus* isolate of interest (Fig. 1c). We previously showed that exogenous T6SS1 E/I pairs can be utilized by the T6SS1 of the RIMD 2210633 isolate to intoxicate a parental RIMD 2210633 that does not carry the cognate immunity[22]. This phenomenon relies on the fact that *V. parahaemolyticus* isolates, and closely related species, carry nearly identical T6SS1 clusters (Supplementary Data 2)[21]. Therefore, we reasoned that RIMD 2210633 T6SS1 can be utilized as a surrogate T6SS platform to screen T6SS1 antibacterial E/I pair candidates from other *V. parahaemolyticus* isolates. True exogenous T6SS1 E/I pairs should provide such a surrogate strain with a T6SS1-dependent competitive advantage over a parental strain lacking the cognate immunity (Fig. 1c).

To establish a RIMD 2210633 T6SS1-based surrogate platform, we first generated a strain in which T6SS1 activity was optimal. To this end, we constructed a RIMD 2210633 derivative with a

deletion in *hns*, which is a negative T6SS1 regulator whose deletion was previously shown to constitutively activate *V. parahaemolyticus* T6SS1[42]. As expected, this mutation resulted in a strain with a constitutively active T6SS1 (Supplementary Fig. 2a–b). We then confirmed the ability of this strain to utilize a validated exogenous T6SS1 E/I pair, V12G01_02265/0 from *V. alginolyticus* 12G01 (a strain that carries a T6SS nearly identical to *V. parahaemolyticus* T6SS1)[22], during T6SS1-mediated competition (Supplementary Fig. 2c). In addition, we tested the surrogate platform using another recently identified *V. parahaemolyticus* 12-297/B T6SS1 E/I pair, B5C30_14465/0[26]. This E/I pair was previously shown to require a cognate VgrG (VgrG1b), which is encoded within the same operon, for T6SS1-mediated delivery[26]. Even with this added complexity, our surrogate platform was able to utilize the E/I pair in T6SS1-mediated competition (Supplementary Fig. 2c). Based on these results, we propose that the RIMD 2210633 Δ*hns* strain can be used as a surrogate platform to screen candidate T6SS1 antibacterial E/I pairs, even when these pairs require an exogenous cognate tail tube component.

**VPBB_RS15030-35 are a T6SS1 E/I pair**. We used the surrogate T6SS1 platform to screen the seven T6SS1 effector candidates that were identified in BB22OP. The seven candidates were cloned together with their neighboring putative immunity genes (Supplementary Data 4) into an arabinose-inducible expression vector. The proteins encoded at the 3′ of the putative E/I cassette (either the putative effector or the putative immunity) were cloned in-frame with a C-terminal 6xHis/Myc tag to allow monitoring the expression of the cassette (Supplementary Fig. 3). These plasmids were introduced into the surrogate platform strains, and their ability to confer T6SS1-mediated killing of a parental RIMD 2210633 prey was determined in a quantitative competition assay, as well as in a qualitative assay[43] by monitoring the survival of GFP-expressing prey. Notably, we confirmed the expression of all but one candidate E/I pair by immunoblot analysis (Supplementary Fig. 3). Our screen revealed that one of the candidate E/I pairs, VPBB_RS15030-35 (accession numbers WP_015297525.1 and WP_005378559.1, respectively), provided the surrogate platform with a T6SS1-dependent ability to kill the parental prey during competition (Fig. 2). This result suggests that VPBB_RS15030-35 are a T6SS1 antibacterial E/I pair. We named them Tme1 (Type VI membrane-disrupting effector 1) and Tmi1 (Type VI membrane-disrupting immunity 1), respectively, as explained below. Notably, any of the other effector candidates could be a true T6SS1 effector that failed to be utilized by the surrogate platform.

**BB22OP T6SSs are functional antibacterial systems**. To validate and characterize BB22OP T6SS1 effectors, we first had to demonstrate that the BB22OP T6SS1 is a functional antibacterial system. Similar to T6SS1 from the studied RIMD 2210633 isolate[40], BB22OP T6SS1 was active under warm marine-like conditions (3% NaCl, 30 °C) in the presence of surface sensing (mimicked in suspension by the addition of the polar flagella inhibitor, phenamil), as evident by secretion of the hallmark T6SS1 component VgrG1 (Supplementary Fig. 4a). However, inactivation of BB22OP T6SS1 by deletion of the conserved structural component, *hcp1*, had only a minor effect on BB22OP's ability to kill *E. coli* prey during competition (Supplementary Fig. 4b). This result suggests that another component contributed to the antibacterial toxicity of BB22OP. Whereas a previous report showed that *V. parahaemolyticus* T6SS2 was inactive under warm marine-like conditions in the studied RIMD 2210633 isolate, we found a strong expression and secretion of the hallmark

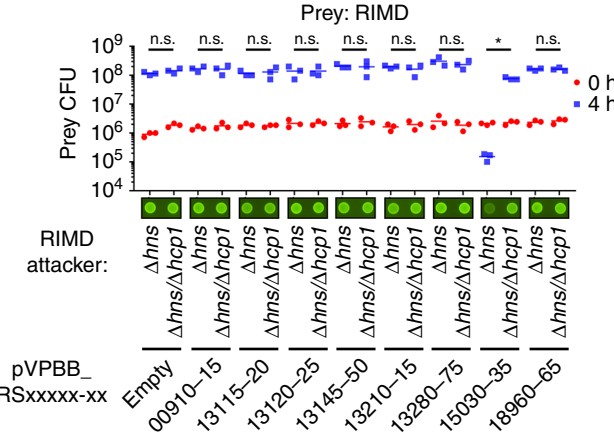

**Fig. 2 Surrogate T6SS1 platform reveals an antibacterial effector in *V. parahaemolyticus* BB22OP.** Viability counts of *V. parahaemolyticus* RIMD 2210633 parental prey before (0 h) and after (4 h) co-incubation with surrogate platform *V. parahaemolyticus* RIMD 2210633 attacker strains, Δ*hns* (T6SS1+) and Δ*hns*/Δ*hcp1* (T6SS1−), on media containing L-arabinose at 30 °C ($n = 3$ co-cultures). Attackers harbor plasmids for the arabinose-inducible expression of the indicated *V. parahaemolyticus* BB22OP proteins (pVPBB_RSxxxx-xx-myc). Asterisk denotes statistical significance between samples at the 4 h timepoint by an unpaired, two-tailed Student's *t*-test ($P = 0.00009$); n.s., no significant difference ($P > 0.05$). Images of representative spots of bacterial co-cultures, in which the indicated attacker strains were mixed with a parental RIMD 2210633 strain constitutively expressing GFP, are shown. Survival of GFP-expressing prey was qualitatively assessed by monitoring GFP fluorescence. Source data are provided as a source data file.

T6SS2 component Hcp2 in BB22OP under these conditions (Supplementary Fig. 4a). Moreover, inactivation of BB22OP T6SS2 by deletion of *hcp2* resulted in a significantly reduced ability to kill *E. coli* prey (Supplementary Fig. 4b). Therefore, we concluded that both T6SS1 and T6SS2 contribute to the antibacterial toxicity of BB22OP.

Since the antibacterial contribution of BB22OP T6SS1 was minor under the tested conditions, we reasoned that deleting *hns*, which was previously shown to be a negative regulator of T6SS1 in the RIMD 2210633 isolate[42], should de-repress T6SS1 activity and allow us to better visualize the T6SS1-mediated phenotypes in BB22OP. As shown in Supplementary Fig. 4a, VgrG1 expression and secretion were activated in BB22OPΔ*hns* even in the absence of surface sensing induction, thus confirming that H-NS is a negative regulator of BB22OP T6SS1. Moreover, deletion of *hcp1* in the Δ*hns* background resulted in a significant loss of toxicity toward *E. coli* prey (Supplementary Fig. 4b). These results indicate that in a Δ*hns* background, BB22OP T6SS1 was a major antibacterial component. Therefore, we decided to use BB22OPΔ*hns* as the parental strain in subsequent bacterial competition experiments, to allow us to better detect the antibacterial activity of BB22OP T6SS1. Notably, deletions of neither *hcp1* nor *hcp2* affected bacterial growth in BB22OP wild-type or Δ*hns* backgrounds (Supplementary Fig. 4c).

**Tme/i1 are an antibacterial BB22OP T6SS1 E/I pair**. Upon determining the conditions required to activate BB22OP T6SS1, we proceeded to validate that Tme/i1 are a T6SS1 E/I pair in BB22OP (Fig. 3a). Indeed, Tme1 expressed from a plasmid was secreted from a T6SS1+ BB22OP strain (BB22OPΔ*hns*), but not from a T6SS1− mutant (BB22OPΔ*hns*Δ*hcp1*) (Fig. 3b). Next, we constructed a BB22OP *tme/i1* deletion strain and used it as prey

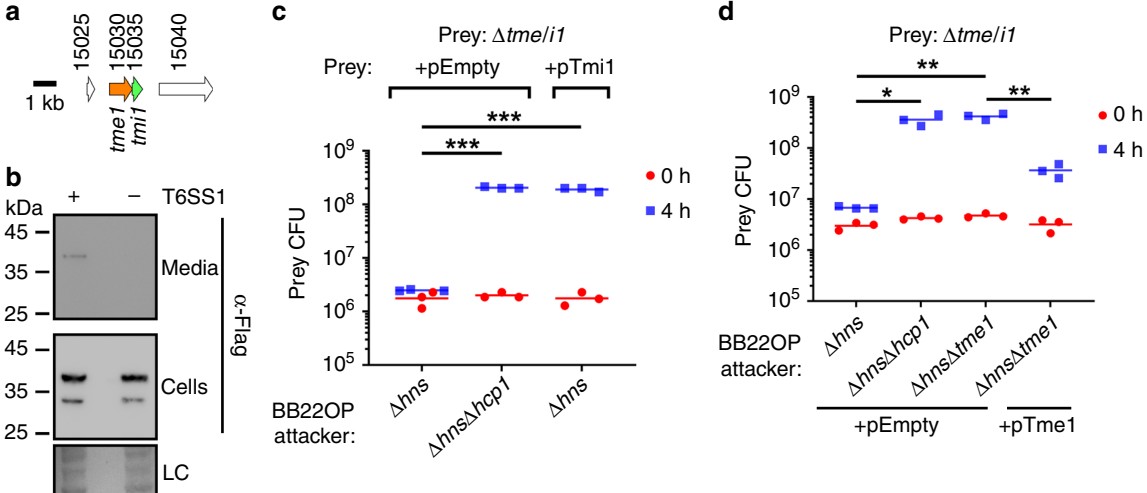

**Fig. 3 Tme/i1 are an antibacterial BB22OP T6SS1 E/I pair. a** Gene structure of *tme/i1*. Genes are represented by arrows indicating the direction of translation. White arrows denote genes unrelated to T6SS. Locus tags (VPBB_RSxxxxx) are shown above. **b** Expression (cells) and secretion (media) of C-terminally Flag-tagged Tme1 from T6SS1+ (Δ*hns*) and T6SS1− (Δ*hns*/Δ*hcp1*) *V. parahaemolyticus* BB22OP strains were detected by immunoblotting using anti-Flag antibodies. Tme1-Flag was expressed from an arabinose-inducible plasmid (pTme1). Loading control (LC), visualized as Ponceau S stained membrane, is shown for total protein lysates. The experiment was independently repeated three times with similar results. Results from a representative experiment are shown. **c**, **d** Viability counts of *V. parahaemolyticus* BB22OP prey before (0 h) and after (4 h) co-incubation with the indicated *V. parahaemolyticus* BB22OP attackers on media containing L-arabinose at 30 °C (*n* = 3 co-cultures). Δ*hns* was used as the parental T6SS1+ strain, and Δ*hns*/Δ*hcp1* was used as a T6SS1− control. In (**c**), the prey contains an empty expression vector (pEmpty) or a vector for the arabinose-inducible expression of Tmi1 (pTmi1). In (**d**), attackers contain an empty expression vector (pEmpty) or a vector for the arabinose-inducible expression of Tme1 (pTme1). Asterisks denote statistical significance between samples at the 4 h timepoint by an unpaired, two-tailed Student's *t*-test (*$P < 0.005$; **$P < 0.0005$; ***$P < 0.00005$). Source data are provided as a source data file.

in self-competition assays. The Δ*tme/i1* mutant lost immunity against a T6SS1+ attacker (Δ*hns*) (Fig. 3c), indicating that one of the deleted genes was necessary to antagonize a T6SS1-mediated attack. Exogenous expression of the predicted immunity protein, Tmi1, from a plasmid (pTmi1) was sufficient to restore immunity against T6SS1-mediated toxicity (Fig. 3c). Tme1 was responsible for the T6SS1-mediated toxicity, since its deletion from the T6SS1+ BB22OP attacker eliminated its ability to intoxicate a Δ*tme/i1* prey (Fig. 3d). Notably, deletion of *tme1* did not affect BB22OP growth (Supplementary Fig. 4c). Furthermore, *tme1* was not required for overall T6SS1 activity, as evident by the unaffected ability of a BB22OP mutant deleted for *tme1* (BB22OPΔ*hns*Δ*hcp2*Δ*tme1*) to kill *E. coli* prey (Supplementary Fig. 4d), as well as by the unaffected secretion of VgrG1 in a Δ*tme1* mutant (Supplementary Fig. 4a). Taken together, the above results indicate that Tme/i1 are a *bona fide* BB22OP T6SS1 antibacterial E/I pair.

**Tme1 belongs to a widespread family of T6SS effectors.** Analysis of Tme1 did not reveal similarity to any known toxin domain or T6SS-associated domain. Therefore, we sought to characterize this effector. Multiple sequence alignment of Tme1 homologs, identified in PSI-BLAST[44], revealed a conserved region at the C-terminus of the protein (corresponding to residues 161–310), which we named Tme. Analysis of Tme domain sequences revealed a conserved motif centered around an almost invariant DxxK, followed by a predicted transmembrane helix that was flanked by another aspartic acid (D) (Fig. 4a).

Analysis of the phylogenetic distribution of Tme domain sequences revealed a diverse family (Fig. 4b). Tme family members were prevalent in Proteobacterial genomes belonging to the alpha-, beta-, gamma-, delta-, and epsilon-proteobacteria classes (Supplementary Data 5). Remarkably, 77% of the Tme-encoding genomes also harbored T6SS (containing at least 9 out of 11 T6SS core components that were shown to specifically

predict T6SS[1]), and 99.7% of the genomes harbored at least one *hcp* gene and one *vgrG* gene, which are hallmark T6SS-secreted components (Supplementary Data 6). Moreover, analysis of Tme-containing proteins revealed that although most harbor N-terminal extensions not similar to any characterized domain (Fig. 4c; "Unknown"), others harbor N-terminal domains that are associated with T6SS, such as PAAR[10], MIX[23], VgrG[2], and ImpA[1] (Fig. 4c and Supplementary Data 5). Only one occurrence of a known N-terminal domain that is not associated with T6SS effectors, but rather with effectors of the type VII secretion system (T7SS), was identified (i.e., LXG)[45]. Taken together, these results suggest that Tme-containing proteins constitute a family of effectors that are predominantly associated with T6SSs. Further support for this conclusion was revealed upon analyzing the genetic neighborhood of Tme-encoding genes; approximately 70% of them had an adjacent upstream gene encoding T6SS-associated proteins (i.e., T6SS core-components, accessory components, adapters, and effectors) (Fig. 4d and Supplementary Data 5).

Next, we characterized the cognate immunity proteins of Tme. Usually, toxin domains are encoded adjacent to their cognate immunity[46]. Since Tme domains were always located at the C-terminus (Fig. 4c), we reasoned that the immunity proteins were encoded downstream. Initial inspection of the predicted immunity proteins (collectively named Tmi, for Type VI membrane-disrupting immunity), which are encoded immediately downstream of Tme family members (Supplementary Data 5), revealed that although most of them are predicted to harbor three to six transmembrane helices, their sequences are considerably diverse. This implies that Tmi proteins cannot be presented as a concise, framed domain as was previously described for other T6SS immunity families. We therefore grouped the predicted Tmi sequences based on all-against-all pairwise similarity (using the CLANS classification tool)[47]. The

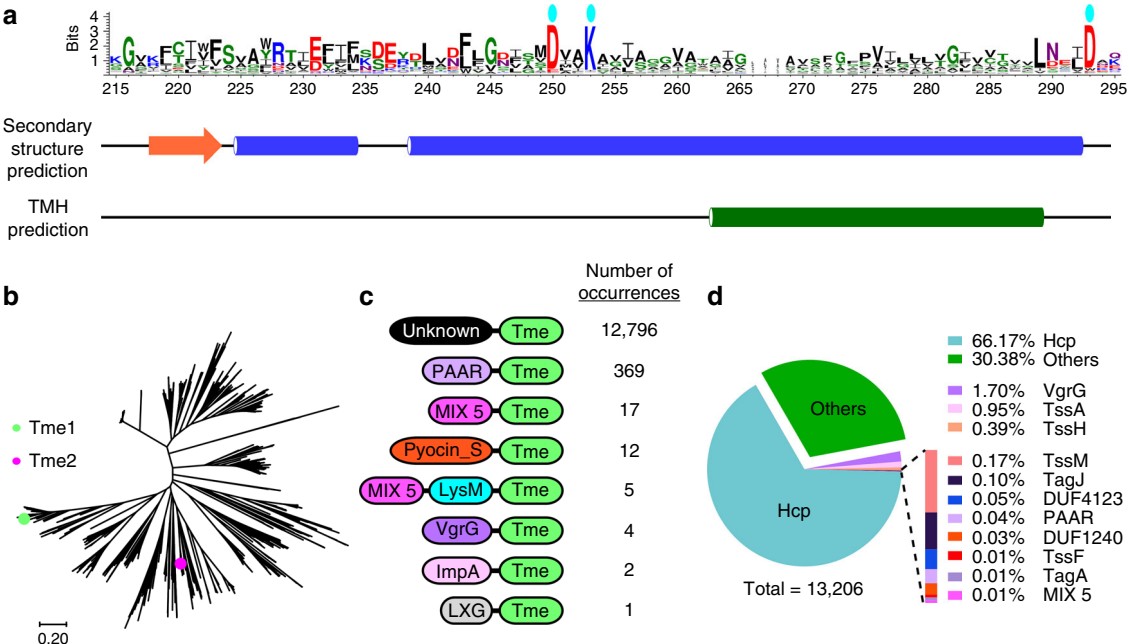

**Fig. 4 Tme1 contains a domain that defines a widespread family of T6SS effectors. a** Conserved motif found in the C-terminal region of Tme1 (Tme) is illustrated using WebLogo 3, based on multiple sequence alignment of Tme1-homologous proteins. Cyan ovals above the WebLogo denote conserved amino acids. Secondary structure prediction (by Jpred) and transmembrane helix (TMH) prediction (by Phobius) are provided below. Alpha helices are denoted by blue cylinders, and a beta strand by an orange arrow. **b** Phylogenetic distribution of Tme domains constructed using the Neighbor-joining method. Tme1 and Tme2 are denoted by green and magenta circles, respectively. The scale bar represents the number of amino acid substitutions per site. **c** Domain architecture of Tme-containing proteins. Domain sizes are not to scale. **d** Pie chart of genes found immediately upstream of Tme-encoding genes. Genes encoding known T6SS structural or accessory components are indicated by their name. "Others" denotes instances where no T6SS-related genes are transcribed in the same direction as the Tme-encoding gene. The percentage of occurrences of each gene is listed next to its name.

resulting similarity map showed that most of the Tmi sequences were indeed connected (Supplementary Fig. 5). Moreover, most clustered groups, including the group containing Tmi1 (Supplementary Fig. 5, green circle), were directly linked to a central node comprising proteins with a domain of unknown function 1240 (DUF1240) (Supplementary Fig. 5, blue circles). Therefore, we propose that the family of proteins that contain a DUF1240 or a DUF1240-like domain can be redefined as Tmi. Notably, several small groups of proteins encoded downstream of Tme members were not linked to any DUF1240-associated nodes. They may represent proteins that were misannotated. Alternatively, they may represent divergent families that also provide immunity against Tme effectors.

***V. parahaemolyticus* T9109 encodes a T6SS1 Tme/i pair**. To further expand our analysis of Tme, we examined an additional family member that was located on a branch of the Tme phylogenetic tree that was different from the one containing Tme1 (Fig. 4b, green and pink circles). We predicted that this Tme-containing protein (PO79_RS05910, accession number WP_047706523.1), encoded by the T6SS1+ (Supplementary Fig. 1) clinical *V. parahaemolyticus* isolate T9109[48], is a T6SS1 effector that is encoded next to its downstream immunity (PO79_RS05915, accession number WP_025509810.1) (Fig. 5a). We noted, based on an analysis of the PO79_RS05910 coding sequence, that the start site of this putative effector gene was probably incorrectly annotated. Therefore, in subsequent analyses we used an open reading frame that extends 36 bases upstream of the NCBI record (see the "Methods" section). We named the T9109 putative E/I pair Tme/i2. Tme2 maintained all of the abovementioned characteristics of a T6SS1 candidate effector, and did not contain identifiable domains

other than Tme. In agreement with our prediction, the *tme/i2* cassette enabled T6SS1-dependent parental killing by the surrogate platform (Supplementary Fig. 6). Therefore, we hypothesized that Tme/i2 are a T6SS1 antibacterial E/I pair in isolate T9109.

The T9109 T6SS1 was active under warm marine-like conditions, as evident by VgrG1 secretion (Supplementary Fig. 7a). Furthermore, T6SS1 was the main cause of antibacterial toxicity under these conditions, since deletion of the conserved *hcp1* resulted in complete loss of *E. coli* prey killing during competition (Supplementary Fig. 7b). Notably, deletion of *hcp1* had no negative effect on T9109 growth (Supplementary Fig. 7c). These results indicate that T9109 T6SS1 is an antibacterial system that is active under warm marine-like conditions. In contrast to BB22OP, we did not use a Δ*hns* background in subsequent T9109 T6SS1 analyses, since it was not required to observe T6SS1-mediated antibacterial activity.

Tme2 was secreted in a T6SS1-dependent manner from T9109, as demonstrated for Tme1 in BB22OP and as expected of a T6SS1 effector (Fig. 5b). To validate that Tme/i2 are a T6SS1 antibacterial E/I pair, we constructed a T9109 *tme/i2* deletion strain and used it as prey in self-competition. As shown in Fig. 5c, the Δ*tme/i2* mutant was no longer able to antagonize T6SS1-mediated attacks due to loss of Tmi2-mediated immunity. Tme2 was the effector required for the T6SS1-mediated toxicity (Fig. 5d). Notably, deletion of *tme2* did not affect T9109 growth (Supplementary Fig. 7c). Furthermore, *tme2* was not required for T6SS1 activity in T9019, as evident by VgrG1 secretion (Supplementary Fig. 7a) and by the killing of *E. coli* prey (Supplementary Fig. 7b) in a Δ*tme2* mutant, which were comparable to those of the parental strain. Taken together, the above results indicate that Tme/i2 are a *bona fide* T9109 T6SS1 antibacterial E/I pair.

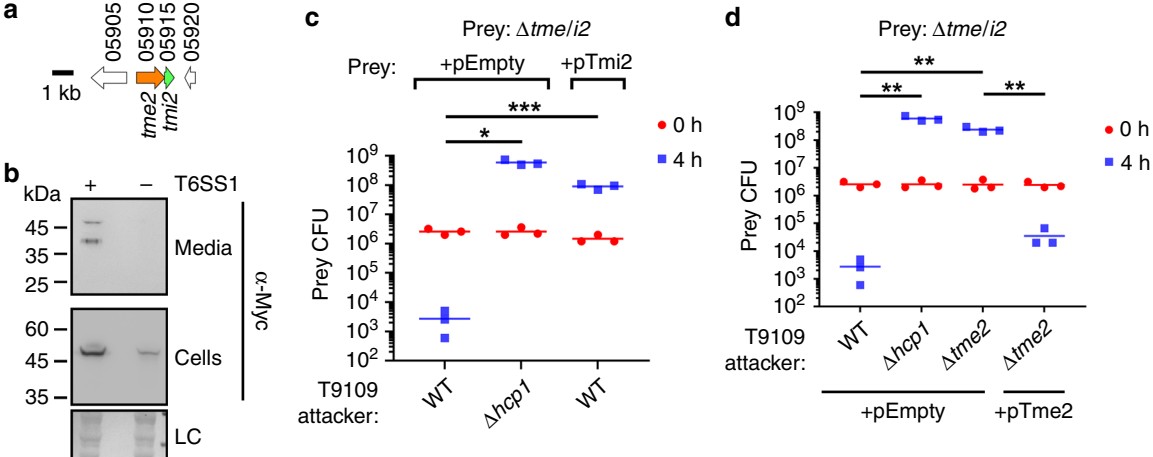

**Fig. 5 Tme/i2 are an antibacterial T9109 T6SS1 E/I pair. a** Gene structure of *tme/i2*. Genes are represented by arrows indicating the direction of translation. White arrows denote genes unrelated to T6SS. Locus tags (PO79_RSxxxxx) are shown above. **b** Expression (cells) and secretion (media) of C-terminally Myc-tagged Tme2 from T6SS1+ (WT) and T6SS1− (Δ*hcp1*) *V. parahaemolyticus* T9109 strains grown in the presence of phenamil (20 μM) were detected by immunoblotting using anti-Myc antibodies. Tme2-Myc was expressed from an arabinose-inducible plasmid (pTme2). Loading control (LC), visualized as trihalo compounds' fluorescence of the immunoblot membrane, is shown for total protein lysates. The experiment was independently repeated three times with similar results. Results from a representative experiment are shown. **c, d** Viability counts of *V. parahaemolyticus* T9109 prey before (0 h) and after (4 h) co-incubation with the indicated *V. parahaemolyticus* T9109 attackers on media containing L-arabinose at 30 °C ($n = 3$ co-cultures). Δ*hcp1* was used as a T6SS1− control. In **c**, the prey contains an empty expression vector (pEmpty) or a vector for arabinose-inducible expression of Tmi2 (pTmi2). In **d**, attackers contain an empty expression vector (pEmpty) or a vector for the arabinose-inducible expression of Tme2 (pTme2). Asterisks denote the statistical significance between samples at the 4 h timepoint by an unpaired, two-tailed Student's *t*-test (*$P < 0.05$; **$P < 0.005$; ***$P < 0.001$). Source data are provided as a source data file.

**Tme effectors disrupt bacterial membranes.** Following our findings showing that Tme-containing proteins are T6SS effectors, we next sought to characterize the antibacterial toxicity mediated by Tme family members. To this end, we expressed Tme1 and Tme2 in the cytoplasm and the periplasm of *E. coli* from an arabinose-inducible expression vector. As shown in Fig. 6, both Tme1 (Fig. 6a) and Tme2 (Fig. 6b) were toxic upon delivery to the periplasm of *E. coli*. In contrast, the expression of Tme1 in the cytoplasm did not affect bacterial growth. Although cytoplasmic expression of Tme2 did result in slower growth, compared with *E. coli* containing an empty expression vector, the periplasmic version of Tme2 was considerably more detrimental. The apparent toxicity of periplasmic Tme2 prior to arabinose induction was probably the result of leaky expression from the Pbad promoter in the absence of glucose-mediated repression. Notably, the expression of the cytoplasmic versions of Tme1 and Tme2 was detected in *E. coli* by immunoblotting, indicating that the differential toxicity was not due to lack of protein expression in the cytoplasm (Supplementary Fig. 8). Co-expression of Tmi proteins antagonized the toxic effect of their respective Tme effectors in the periplasm (Fig. 6c, d). Notably, the Tmi proteins appear to provide specific immunity against their cognate Tme, since they were unable to provide cross-protection (Fig. 6c, d).

The above findings indicated that Tme effectors are functional upon delivery to the periplasm. Importantly, their activity did not result in cell lysis, since we did not detect any decrease in the culture's turbidity (Fig. 6a–d), suggesting that Tme effectors probably do not hydrolyze the peptidoglycan layer. Taken together with our observation that the Tme domain contains a predicted transmembrane helix (Fig. 4a), this result led us to hypothesize that Tme effectors target the bacterial membrane, possibly by disrupting membrane integrity. This activity was previously associated with several T6SS effectors that function in the periplasm.

Membrane disruption should result in dissipation of the membrane potential and in ion leakage. Therefore, we first examined the effect of periplasmic Tme effectors on *E. coli* membrane potential. To this end, we used the BacLight Bacterial Membrane Potential Kit, which utilizes the fluorescent membrane potential indicator dye, DiOC$_2$(3)[49]. The fluorescence emission of DiOC$_2$(3) is green, but in healthy cells containing a normal membrane potential the dye concentrates and self-associates, causing the emission to shift to red. Consequently, monitoring the red/green fluorescence ratio provides an indication of the membrane potential; a low ratio is typical of cells in which the membrane potential has been dissipated. Remarkably, *E. coli* expressing the periplasmic versions of Tme1 and Tme2 presented a significantly lower red/green ratio, compared with cells containing an empty expression vector (pEmpty). The reduced ratio was similar to that found in cells expressing a periplasmic version of VasX, a previously described pore-forming T6SS effector from *V. cholerae*[49,50], and to that found in cells pre-treated with CCCP, a proton ionophore that eradicates the proton gradient and thus eliminates the membrane potential (Fig. 6e and Supplementary Fig. 9a). This result indicates that the activity of Tme effectors leads to dissipation of membrane potential.

Next, we aimed to determine whether Tme effectors affect the permeability of the membrane. To this end, we used propidium iodide (PI), a membrane-impermeable intercalating dye that fluoresces upon binding DNA. PI fluorescence is indicative of membrane disruption that allows the dye to enter the cell and bind DNA. *E. coli* expressing the periplasmic versions of Tme1 and Tme2 exhibited a significantly higher level of PI fluorescence, compared with cells containing an empty expression vector (pEmpty) (Fig. 6f and Supplementary Fig. 9b). The Tme-mediated increase was similar to that observed in cells expressing a periplasmic version of VasX, and to cells pre-treated with ethanol to permeabilize the membrane. This result indicates that

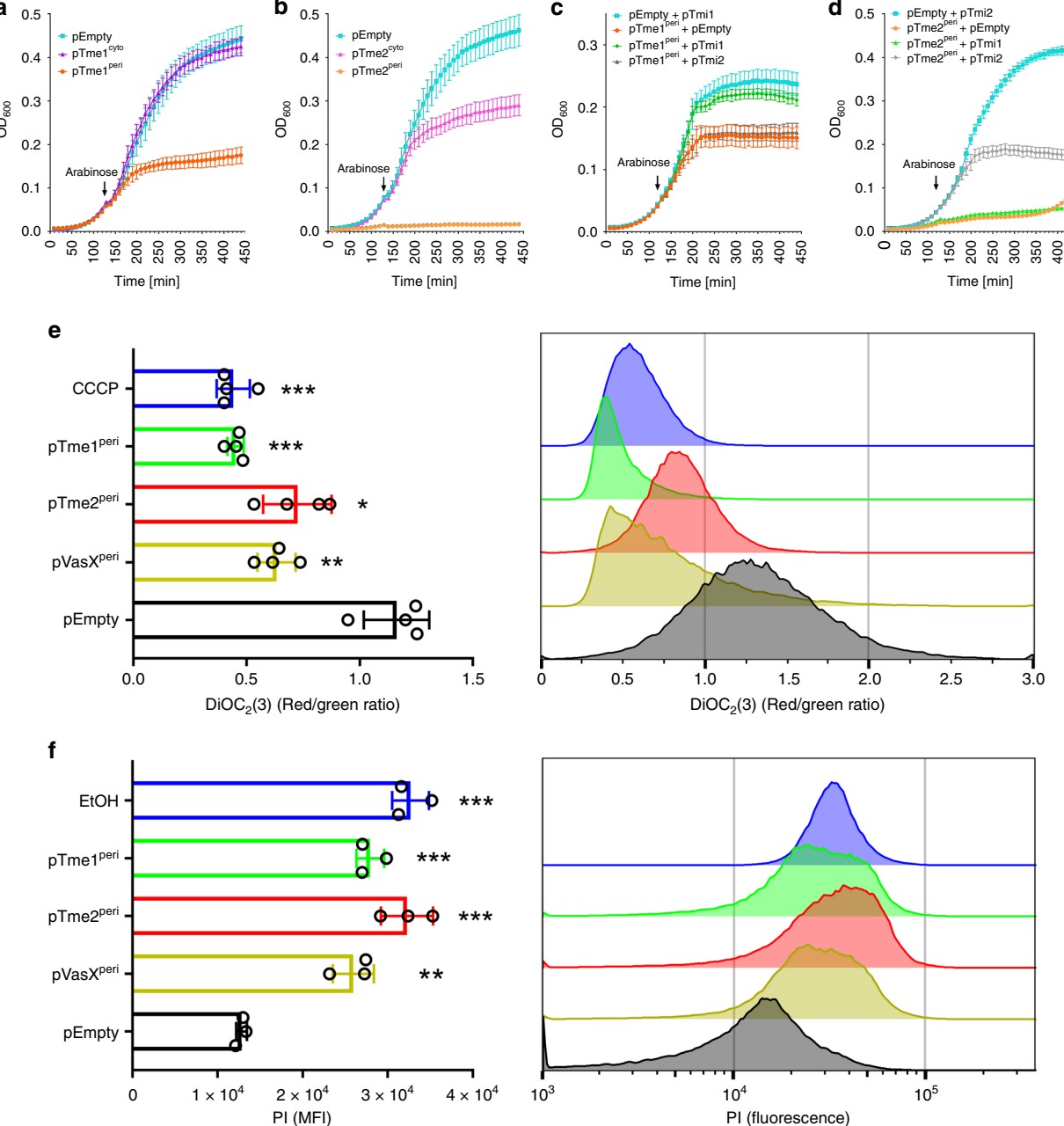

**Fig. 6 Tme effectors disrupt membrane integrity upon delivery to the periplasm. a**, **b** Toxicity of Tme in bacteria. Growth of *E. coli* BL21 (DE3) containing arabinose-inducible vectors for the expression of cytoplasmic (cyto) or periplasmic (peri) versions of Tme1 (**a**) and Tme2 (**b**). **c**, **d** Rescue of Tme-mediated toxicity by cognate Tmi. Growth of *E. coli* BL21 (DE3) containing an arabinose-inducible vector for the expression of periplasmic Tme1 (**c**) or Tme2 (**d**) together with an arabinose-inducible vector for the expression of Tmi1 or Tmi2. Empty expression vectors were used as the control. In **a–d**, data represent the mean ± S.D. (*n* = 4). Arrows denote the time at which L-arabinose was added. **e** Tme effectors dissipate membrane potential. *E. coli* BL21 (DE3) expressing periplasmic Tme1, Tme2, or the *V. cholerae* pore-forming effector VasX (used as a positive control) from arabinose-inducible vectors, were analyzed using flow cytometry following staining using the BacLight Membrane Potential Kit. The red/green fluorescence ratio of the dye $DiOC_2(3)$ was calculated for each condition. CCCP was used as a positive control. Data shown in the left panel represent the mean ± S.D. of four independent experiments. Asterisks denote statistical significance compared to pEmpty samples by one-way repeated measures ANOVA with the Dunnett test (*$P <$ 0.01; **$P <$ 0.001; ***$P <$ 0.0005). Data shown in the right panel represent the distribution of red/green ratios for cells analyzed in one of the experiments shown on the left. **f** Tme effectors increase membrane permeability. *E. coli* cultures, like in (**e**), were stained with the membrane-impermeable, intercalating DNA dye propidium iodide (PI) and analyzed using flow cytometry. Pre-treatment with ethanol (EtOH) was used as a positive control. Data shown in the left panel represent the geometric mean fluorescence intensity (MFI) ± S.D. of three repeats from a representative experiment. Asterisks denote statistical significance compared to pEmpty samples by one-way repeated measures ANOVA with the Dunnett test (**$P <$ 0.001; ***$P <$ 0.0005). Data shown in the right panel represent the distribution of PI fluorescence intensities for cells analyzed in one repeat of the experiment shown on the left. Source data are provided as a source data file.

Tme effectors disrupt bacterial membranes, allowing PI to enter the cells. Taking the above results together, we conclude that Tme effectors target the periplasm of Gram-negative bacteria, causing membrane disruption and loss of membrane potential (hence they were named Tme, for Type VI membrane-disrupting effector).

## Discussion

In this work, we demonstrated the use of a methodology comprising a comparative genomic analysis and a surrogate T6SS screening platform to identify T6SS effectors. Utilizing this methodology to analyze the genome of the *V. parahaemolyticus* isolate BB22OP, we revealed a widespread family of antibacterial T6SS effectors. These effectors function in the bacterial periplasm where they disrupt membrane integrity. Thus, we named this toxin family Tme (Type VI membrane-disrupting effector). Other membrane-disrupting T6SS effectors have been described previously. It was suggested that such effectors (e.g., *V. cholerae* VasX[49,50] and *Pseudomonas aeruginosa* Tse4[51]) form pores in the bacterial membrane. Importantly, however, the conserved motif and the predicted secondary structure of Tme members indicated that they are distinct from previously reported toxins. The vast majority of Tme domains included a predicted transmembrane helix (according to the Phobius transmembrane topology predictor[52]). Notably, manual inspection of the approximately 130 Tme-containing proteins for which no transmembrane helix was detected revealed that they possess a putative transmembrane helix that was slightly below the predictor threshold value. Such a transmembrane helix might play a role in disrupting membrane integrity.

Tme domains are widespread and are encoded by bacteria belonging to all major Proteobacterial classes. Unlike many other toxin families (e.g., Tne2[53] and PoNe[26]) that are found fused to the delivery domains of various antibacterial toxin secretion systems, Tme domains appear to be almost exclusively restricted to T6SS effectors. This conclusion was based on several pieces of evidence. First, we experimentally confirmed that Tme1 and Tme2, located on distinct branches of the Tme phylogenetic tree, are antibacterial T6SS effectors. In addition, 77% of Tme-encoding genomes harbor a T6SS, representing a significant enrichment compared with only ~25% of bacterial genomes that are predicted to harbor T6SS[54]. Moreover, 99.7% of Tme-containing proteins are encoded in genomes that harbor at least one *hcp* and one *vgrG* genes, which are hallmark T6SS-secreted tail tube components. Tme-containing proteins are also encoded immediately downstream of a T6SS-related gene in approximately 70% of the cases. Furthermore, the known domains that we identified at the N-termini of Tme-containing proteins were associated with T6SS delivery (e.g., PAAR[10], VgrG[2], and MIX[23]), but not with other antibacterial toxin secretion systems. One exception was a Tme-containing protein that was identified in a *Bacillus cavernae* genome (WP_126867112.1); it was fused to the T7SS delivery domain LXG[45]. Nevertheless, phylogenetic analysis revealed that this LXG-fused member was located on an isolated, distinct branch of the Tme family tree.

We also noted that Tme was found fused to an N-terminal ImpA domain (in WP_079984663.1 and WP_080209522.1). The ImpA domain was recently analyzed and shown to be mainly present in two T6SS structural components, TssA (e.g., when fused to C-terminal VasJ domain) and TagA (e.g., when fused to C-terminal VasL domain)[55,56]. Both TssA and TagA play a role in the biogenesis of the T6SS tube by terminating its elongation. Therefore, it is intriguing to find ImpA domains fused to a C-terminal Tme toxin domain. This finding raises the possibility that ImpA domains play several roles in the T6SS context, not only as structural domains involved in T6SS biogenesis but also as a delivery domain for effectors. The identity of these ImpA-fused Tme proteins as antibacterial toxins is further supported by the presence of a downstream gene that encodes a protein containing a DUF1240-like domain (according to HHpred[57] analysis), which we established as the main immunity family of Tme (see below).

Tme effectors appear to be linked to a rather diverse family of immunity proteins, which we collectively named Tmi. The majority of proteins encoded immediately downstream of the Tme effectors are predicted to contain transmembrane helices, which are expected from proteins that provide immunity against a membrane-disrupting toxin. Notably, while most Tmi proteins do not contain N-terminal signal peptides, they do contain N-terminal transmembrane helices that may serve as membrane anchors[58,59] (Supplementary Data 5). Although the variance in the primary sequences of Tmi proteins prevented us from identifying a coherent immunity domain using standard multiple sequence alignment, an all-against-all pairwise similarity analysis revealed that most Tmi proteins were connected to a central node of the DUF1240-containing proteins. Indeed, the experimentally validated immunity proteins Tmi1 and Tmi2, which were located on separate nodes in this analysis and shared only a low level of sequence similarity, were both directly linked to the DUF1240 node. Moreover, homology detection and structure prediction by the HHpred tool[57] detected similarity (probability > 95%) to DUF1240 in the sequences of both Tmi1 and Tmi2, suggesting that they are DUF1240-like proteins. Therefore, we concluded that DUF1240 and DUF1240-like proteins provide immunity against Tme effectors. Notably, additional proteins encoded downstream of Tme effectors were not linked to the DUF1240 family; they may constitute distinct Tme immunity families. Interestingly, an association with a diverse array of immunity partners that possess varying membrane topologies had been previously reported for the peptidoglycan-targeting Colicin M toxin family[60], suggesting that different immunity mechanisms had evolved to antagonize its toxicity. It is reasonable to assume that although DUF1240 proteins constitute the main Tme immunity family, additional families and immunity mechanisms have evolved to antagonize Tme-mediated toxicity. Future studies are required to decipher the biochemical and biophysical properties of Tme domains, their structure, and the mechanisms underlying the antagonistic effect of Tmi family members against them.

The methodology that we employed here to reveal a T6SS1 effector in the *V. parahaemolyticus* BB22OP genome comprised two stages. The first was a computational approach based on comparative genomics; it relied on the premise that T6SS effectors are genetically linked to the system through which they are delivered, and that they are not present in genomes in which the system is absent. The second stage was the surrogate T6SS platform, which provided a fast and easy assay to screen antibacterial T6SS1 E/I pairs from a list of potential candidates. The surrogate platform was developed based on prior observations, which indicated that T6SS effectors are interchangeable between strains that harbor nearly identical T6SSs (as is the case for *V. parahaemolyticus* T6SS1).

Although we used our methodology to analyze only one *V. parahaemolyticus* genome in this report, this pipeline can be easily scaled up to simultaneously survey the genomes of hundreds of *V. parahaemolyticus* isolates to search for T6SS1 effectors. Furthermore, this methodology is not limited to *V. parahaemolyticus*, and can also be employed to identify effectors of T6SSs in other bacterial genera. However, for this methodology to be applicable for any bacterial species of interest, a large genomic dataset of T6SS+ and T6SS− isolates should ideally be available, and the T6SS in question should be highly similar

between the different isolates. A single isolate harboring the T6SS of interest should be generated, in which T6SS can be activated to serve as a surrogate platform to screen candidate effectors that were identified in other isolates during the comparative genomic stage.

Importantly, as shown here with Tme, the discovery of a novel effector in one strain can lead to uncovering widespread families of toxins that are used by diverse bacteria. Whereas Tme was found to be predominantly associated with T6SS, other toxin domains that will be discovered using this methodology may be associated with diverse delivery mechanisms, and may thus have broad implications on and advance our understanding of bacterial warfare and toxic arsenals.

Noteworthy, the development of a surrogate T6SS platform is pivotal for the future scale up of this methodology; it allows one to screen effector candidates from any isolate that harbors the T6SS in question, even from isolates that are not amenable for genetic manipulations. Moreover, the use of our previously developed methodology[43], in which prey survival is qualitatively assessed by monitoring GFP levels expressed by the prey strain, obviates the need for laborious quantitative competition assays as an initial screen of candidates. To test a candidate effector in the surrogate platform, one simply needs to amplify the DNA sequence of the candidate E/I pair in question (or order the cassette from a variety of commercial vendors), introduce it into the surrogate platform strain (and into its T6SS⁻ negative control derivative), and employ it in bacterial competition (either quantitative or qualitative) against the surrogate platform's parental strain that lacks immunity against the exogenous effectors. Therefore, such a surrogate platform negates the need to construct deletion strains and perform self-competition assays for each isolate that harbors a candidate effector, and it provides a single platform enabling a quick and easy screen for T6SS E/I pairs.

Considering the above characteristics, our methodology has several advantages over other unbiased approaches that have been used to date in the T6SS field: (i) it allows one to screen for effectors even in the genomes of isolates that are not available in the laboratory, or that are not culturable; (ii) it does not require one to genetically manipulate all isolates of interest; (iii) it requires no prior knowledge of the conditions required to activate the T6SS in multiple isolates. In conclusion, in this work we have established a methodology, relying on a comparative genomics approach that complements the available bioinformatic and experimental approaches aimed at identifying T6SS effectors. Scaling this methodology up to simultaneously analyze dozens of genomes will undoubtedly result in the identification of additional effectors.

## Methods

**Strains and media.** *Vibrio parahaemolyticus* isolates RIMD 2210633[61], BB22OP[41], T9109 (a gift from Swapan Banerjee[48]), and their derivatives were grown in MLB media (Lysogeny broth media containing 3% wt/vol NaCl) or on marine minimal media (MMM) agar plates (1.5% wt/vol agar, 2% wt/vol NaCl, 0.4% wt/vol galactose, 5 mM MgSO₄, 7 mM K₂SO₄, 77 mM K₂HPO₄, 35 mM KH₂PO₄, and 2 mM NH₄Cl) at 30 °C. In cases where *V. parahaemolyticus* contained a plasmid, chloramphenicol (10 μg/mL), kanamycin (250 μg/mL), streptomycin (100 μg/mL), or gentamycin (50 μg/mL) was added to maintain the plasmid.

*Escherichia coli* DH5α were used for competition assays (see below) and for plasmid construction and maintenance. *E. coli* DH5α (λ pir) (a gift from Eric V. Stabb) were used for maintaining pDM4 OriR6k suicide vectors and for mating (see below), and *E. coli* BL21 (DE3) were used for protein expression. *E. coli* were grown in 2xYT broth (1.6% wt/vol tryptone, 1% wt/vol yeast extract, and 0.5% wt/vol NaCl) at 37 °C. In cases where *E. coli* contained a plasmid, chloramphenicol (10 μg/mL) or kanamycin (30 μg/mL) was added to maintain the plasmid. When expression of genes from an arabinose-inducible promoter was required, L-arabinose was added to media at 0.05–0.1% wt/vol, as specified.

**Plasmid construction.** For arabinose-inducible expression, the coding sequences (CDS) of putative T6SS1 effectors or immunity genes were amplified from genomic DNA of the relevant *V. parahaemolyticus* isolate. PCR products were inserted into plasmids' multiple cloning site (MCS) using the Gibson-assembly method[62], unless otherwise specified.

To express E/I candidate pairs in the RIMD 2210633-derived surrogate platform, the gene pairs listed in Supplementary Data 4 were amplified from the genome of *V. parahaemolyticus* BB22OP and inserted into the MCS of the pBAD/ *Myc*–His (hereafter named pBAD) vector (Invitrogen) harboring a kanamycin-resistance cassette[40]; the gene found at the 3′ end of the amplified cassette was cloned in-frame with the C-terminal *Myc*-6xHis tag to allow the expression of the cassette to be detected. For surrogate platform validation, the genes encoding the T6SS1 E/I pair, B5C30_RS14465 and B5C30_RS14460, from *V. parahaemolyticus* 12-297/B[26] were amplified with or without the upstream *vgrG1b* gene, and inserted into pBAD as detailed above. Construction of pBAD for the arabinose-inducible expression of the *V. alginolyticus* 12G01 T6SS1 E/I pair, V12G01_02265 and V12G01_02260, was described previously[22].

For cytosolic expression of effectors in *E. coli*, genes were inserted into pBAD, in-frame with the C-terminal *Myc*-6xHis tag. For periplasmic expression, genes were inserted into the Kan^R pPER5/*Myc*-His vector[17] (hereafter named pPER5), a pBAD derivative in which the PelB signal sequence was inserted at the 5′ of the multiple cloning site. The accession numbers of the amplified effectors were WP_015297525.1 (Tme1) from isolate BB22OP, WP_047706523.1 (Tme2) from isolate T9109, and NP_232421.1 (VasX), a *Vibrio cholerae* pore-forming effector from *V. cholerae* V52. For *tme2*, we noted upon inspection of the annotated CDS, and after analysis using the gene recognition and translation initiation site identifier Prodigal[63], that the correct start site is found 36 bases upstream of the NCBI-annotated start site; therefore, we used this revised CDS as *tme2* (positions 165330–166532 of the NCBI reference sequence NZ_JTGR01000043.1). For complementing the *tme1* deletion mutant, and for secretion assessment of Tme1 from BB22OP, *tme1* was cloned into the MCS of pBAD33.1 (Addgene) in-frame with a C-terminal Flag-tag. For complementing the *tme2* deletion strain, and for secretion assessment of Tme2 from T9109, the abovementioned pBAD, in which Tme2 was cloned in-frame with a C-terminal *Myc*-6xHis, was used.

For arabinose-inducible expression of immunity proteins, genes were inserted into the MCS of pBAD33.1 (Addgene) in which the sequence encoding a Flag-tag was inserted at the 3′ of the MCS. The accession numbers of the immunity proteins are WP_005378559.1 (Tmi1) from isolate BB22OP and WP_025509810.1 (Tmi2) from isolate T9109.

Primers used in this study are listed in Supplementary Table 1.

The constructed plasmids were transformed into *E. coli* competent cells using electroporation or using the Mix & Go! *E. coli* transformation kit (Zymo Research), according to the manufacturer's instructions. Transformants containing a plasmid with the correct insert were identified by PCR using the primer pair for pBAD vectors, and were then confirmed by sequencing. During plasmid construction, 0.2% wt/vol glucose was added to the culture media (broth or plates) to repress the expression of the putative effectors from the *Pbad* promoter. To introduce plasmids into *V. parahaemolyticus*, the relevant plasmids were conjugated into *V. parahaemolyticus* using tri-parental mating. Trans-conjugants were selected on MMM agar plates supplemented with appropriate antibiotics.

**Construction of deletion strains.** For in-frame deletions in *V. parahaemolyticus* isolates, 1 kb sequences upstream and downstream of each gene or operon to be deleted were cloned into pDM4, a Cm^R OriR6K suicide plasmid[64]. These pDM4 constructs were transformed into *E. coli* DH5α (λ pir) by electroporation, and then transferred into *V. parahaemolyticus* isolates using conjugation. Trans-conjugants were selected on MMM agar plates containing chloramphenicol (10 μg/mL). The resulting trans-conjugants were grown on MMM agar plates containing sucrose (15% wt/vol) for counter-selection and loss of the SacB-containing pDM4. Deletions were confirmed by PCR. A pDM4 construct for deletion of *hns* in *V. parahaemolyticus* RIMD 2210633 was reported previously[42]. Construction of *V. parahaemolyticus* RIMD 2210633 Δ*tdhAS*/Δ*vp1415-6* was reported previously[23], as was construction of *V. parahaemolyticus* RIMD 2210633 Δ*hcp1*[26].

**Quantitative bacterial competition assays.** Quantitative bacterial competition assays were performed as previously described[40]. *E. coli* DH5α prey harbored a pBAD33 (Addgene) plasmid, providing selectable resistance against chloramphenicol. *V. parahaemolyticus* prey harbored a pBAD33 plasmid (BB22OP, T9109, and RIMD 2210633) or pBAD18 (Addgene) plasmid (BB22OP), providing selectable resistance against chloramphenicol and gentamycin, respectively. L-arabinose (0.02% or 0.05% wt/vol) was added to assay plates when expression from arabinose-inducible plasmids was required. Assays were repeated at least three times with similar results. Results from a representative experiment are shown.

**Qualitative bacterial competition assays.** Qualitative bacterial competition assays were performed as previously described[43], with minor changes. In brief, surrogate platform strains carrying the indicated E/I candidates were grown overnight in 96-well plates containing 100 μL of MLB supplemented with kanamycin (250 μg/mL).

The next morning, 25 µL of an overnight-grown culture of a parental RIMD 2210633 strain carrying a plasmid for constitutive expression of GFP[65] and an empty pBAD vector to provide kanamycin-resistance were added to each well. Mixed cultures were spotted (5 µL) on MLB plates supplemented with kanamycin (250 µg/mL) to maintain the plasmids, and 0.05% (wt/vol) L-arabinose was used to induce candidate E/I expression. After 5 h, the survival of GFP-expressing prey was assessed using a Fusion FX6 imager (Vilber) equipped with a GFP filter.

**Protein secretion assays**. For VgrG1 (a hallmark secreted protein for T6SS1) and Hcp2 (a hallmark secreted protein for T6SS2) secretion, *V. parahaemolyticus* isolates were grown overnight in MLB. Cultures were normalized to $OD_{600} = 0.18$ in 5 mL MLB and grown for 5 h at 30 °C. Phenamil (20 µM) was added where indicated to induce surface sensing (phenamil is an inhibitor of the polar flagella; it mimics surface sensing activation[40]). After 5 h, for expression fractions (cells), 1 $OD_{600}$ units were collected. Cell pellets were resuspended in (2×) Tris-Glycine SDS Sample Buffer (Novex, Life Sciences). For secretion fractions (media), 10 $OD_{600}$ units were filtered (0.22 µm) and proteins were precipitated from the media with deoxycholate and trichloroacetic acid[66]. Cold acetone was used to wash the protein pellets twice. Air-dried protein pellets were resuspended in 20 µL of 10 mM Tris–HCl pH = 8.0, followed by the addition of 20 µL of (2×) Tris-Glycine SDS Sample Buffer. Expression and secretion samples were boiled for ten or five minutes, respectively, at 95 °C, and then loaded onto TGX stain-free gels (Bio-Rad) for SDS-polyacrylamide gel electrophoresis (SDS-PAGE). Transfer onto nitrocellulose membranes was performed using Trans-Blot Turbo Transfer (Bio-RAD). Custom-made anti-VgrG1 or anti-Hcp2 antibodies (at 1:1000 dilution)[21] were used for immunoblotting. Protein signals were visualized using enhanced chemiluminescence (ECL).

The same protocol was also used to detect the secretion of Tme1 and Tme2 tagged with C-terminal Flag (Tme1) or Myc (Tme2) tags, with minor changes. *V. parahaemolyticus* BB22OP and T9109 strains were grown overnight in MLB with appropriate antibiotics to maintain the Tme expression plasmid. Cultures were normalized to $OD_{600} = 0.18$ in 5 mL MLB supplemented with appropriate antibiotics and 0.05% wt/vol L-arabinose to induce the expression from the plasmid. Normalized cultures were grown for 5 h at 30 °C. Phenamil (20 µM) was added to T9109 where indicated. After 5 h, cells were treated as described above for VgrG1 and Hcp2 secretion. Media fractions were resuspended in Tris-Glycine SDS Sample Buffer supplemented with 5% (vol/vol) β-mercaptoethanol, and 0.5 µL of 1 M NaOH were also added to the lysates to adjust the pH. Expression and secretion samples were boiled for ten or five minutes, respectively, at 95 °C, and then loaded onto TGX stain-free gels (Bio-Rad) or ExpressPlus™ PAGE Gels (GenScript) for SDS-PAGE. Transfer onto nitrocellulose membranes was performed using Trans-Blot Turbo Transfer (Bio-RAD). Membranes were probed with anti-c-Myc antibodies (9E10, sc-40, Santa Cruz Biotechnology) or anti-FLAG (DYKDDDDK Tag, D6W5B, Rabbit mAb #14793, Cell Signaling Technology) at 1:1000 dilution. Protein signals were visualized using ECL. Assays were repeated at least three times with similar results. Results from a representative experiment are shown.

**Vibrio growth assays**. Overnight-grown cultures of *V. parahaemolyticus* were normalized to $OD_{600} = 0.01$ in MLB media, and then transferred in triplicate or quadruplicate to 96-well plates (200 µL per well). Cultures were grown at 30 °C in a BioTek SYNERGY H1 microplate reader with continuous shaking at 205 cpm. $OD_{600}$ readings were acquired every 10 min. Assays were repeated at least three times with similar results. Results from a representative experiment are shown.

***E. coli* toxicity assays**. To examine the toxic effects of Tme1 and Tme2, pBAD and pPER5 plasmids, either empty or harboring the indicated gene, were transformed into *E. coli* BL21 (DE3). Transformants were grown overnight in 2xYT supplemented with kanamycin and 0.2% (wt/vol) glucose (to repress leaky expression from the *Pbad* promoter). Cultures were washed (to remove residual glucose) and normalized to $OD_{600} = 1$ in 2xYT. Cultures were further diluted to $OD_{600} = 0.01$ in 2xYT containing kanamycin (30 µg/mL), and 200 µL of each sample were transferred into 96-well plates in quadruplicate. $OD_{600}$ readings were taken every 10 min for 7 h while plates were grown at 37 °C with agitation (205 cpm) in a microplate reader (BioTek SYNERGY H1). After 2 h of growth, L-arabinose was added to each well to a final concentration of 0.1% (wt/vol), in order to induce expression from the plasmids. Assays were repeated at least three times with similar results. Results from a representative experiment are shown.

To assess the protection conferred by Tmi1 and Tmi2 against Tme1 and Tme2, L-arabinose-inducible pBAD33.1 expression vectors, either empty or harboring the indicated Tmi, were co-transformed with the pPER5 effector-expression plasmids (pTme1peri and pTme2peri) into *E. coli* BL21 (DE3). Transformants were treated as described above with the addition of chloramphenicol (10 µg/mL) to the media to maintain pBAD33.1 plasmids. Assays were repeated at least three times with similar results. Results from a representative experiment are shown.

**Protein expression**. To verify the expression of E/I candidates in the *Vibrio* surrogate platform, the indicated *V. parahaemolyticus* RIMD 2210633 mutants containing vectors for the arabinose-inducible expression of the indicated gene cassettes were grown overnight in MLB supplemented with kanamycin. Cultures were then normalized to $OD_{600} = 0.5$ in MLB supplemented with kanamycin and 0.05% (wt/vol) L-arabinose to induce protein expression. Cultures were grown for 3 h at 30 °C.

To verify the expression of Tme1 and Tme2 in *E. coli*, the abovementioned *E. coli* BL21 (DE3), transformed with pBAD and pPER5 plasmids for the expression of Tme1 or Tme2 (pTme1cyto and pTme1peri, or pTme2cyto and pTme2peri, respectively), were grown overnight in 2xYT supplemented with 30 µg/mL kanamycin and 0.2% (wt/vol) glucose (to repress expression from the *Pbad* promoter). Overnight cultures were then washed to remove residual glucose, and diluted 100-fold into 5 mL of fresh 2xYT media supplemented with antibiotics. Cultures were grown for two additional hours at 37 °C. To induce protein expression, 0.1% (wt/vol) L-arabinose was added to the media, and cultures were incubated at 37 °C for two additional hours.

Next, 0.5 $OD_{600}$ units of induced *E. coli* or *Vibrio* cultures were pelleted and resuspended in a (2×) Tris-Glycine SDS Sample Buffer supplemented with 5% (vol/vol) β-mercaptoethanol, followed by boiling at 95 °C for ten minutes. Samples were resolved on ExpressPlus™ PAGE Gels (GenScript) or TGX stain-free gels (Bio-Rad), and then transferred onto nitrocellulose membranes. Immunoblotting was performed with anti-c-Myc antibodies (9E10, sc-40, Santa Cruz Technology) at 1:1000 dilution. Protein signals were visualized by ECL.

**Membrane potential assays**. To determine whether effectors caused the dissipation of the membrane potential, the BacLight Bacterial Membrane Potential Kit (Molecular Probes, Invitrogen) was used. *E. coli* BL21 (DE3) harboring pPER5 plasmids for expressing the periplasmic forms of the indicated effectors were grown overnight in 2xYT supplemented with 30 µg/mL kanamycin and 0.2% (wt/vol) glucose. Overnight cultures were washed to remove residual glucose, then normalized to $OD_{600} = 0.5$ in 5 mL 2xYT supplemented with kanamycin, and then grown for 2 h at 37 °C with agitation (220 rpm). After 2 h, L-arabinose was added to each culture to a final concentration of 0.05% (wt/vol) to induce expression from plasmids. The cultures were grown for two additional hours at 37 °C with agitation. Induced cultures were divided into triplicates, washed twice with 1 mL of filter-sterilized phosphate-buffered saline (PBS), and resuspended in PBS to $OD_{600} = 0.5$. Samples were then transferred to a Greiner U-shaped 96-well plate (200 µL per well). Then, ten µL DiOC$_2$(3) (3,3'-diethyloxacarbocyanine iodide) were added to all wells to a final concentration of 3 µM. As a positive control, CCCP (carbonyl cyanide 3-chlorophenylhydrazone) was added to a final concentration of 500 µM to *E. coli* containing an empty expression vector (pEmpty), five minutes prior to DiOC$_2$(3) staining. Stained bacteria were incubated for 30 min in the dark at room temperature. Samples were then analyzed using ThermoFisher Scientific Attune NxT flow cytometry. A minimum of 50,000 bacteria were first gated using forward and side scatter. DiOC$_2$(3) was excited using Blue laser (488 nm) and emission was detected using the 530/30 (green) and 590/40 (red) filters. Red/green ratios of gated bacteria were calculated from the geometric mean florescence intensity (MFI) of each channel using FlowJo V10 software.

**Membrane permeability assays**. To determine whether effectors caused increased membrane permeability, *E. coli* BL21 (DE3) cultures were grown and treated as described in the membrane potential assay. As a positive control, *E. coli* containing an empty expression vector (pEmpty) were incubated in 1 mL of 70% ethanol for 15 min, and then washed twice in 1 mL filter-sterilized PBS. Samples were transferred to Greiner U-shaped 96-well plates (200 µL per well). Then, ten µL of Propidium iodide (PI), a membrane-impermeable DNA intercalating dye (Sigma), were added to each well to a final concentration of 1 µg/mL. Stained bacteria were incubated for 15 min in the dark at room temperature. Samples were then analyzed using ThermoFisher Scientific Attune NxT flow cytometry. A minimum of 50,000 bacteria were first gated using forward and side scatter. PI was excited using Yellow laser (561 nm) and emission was detected using the 585/15 filter. Geometric mean florescence intensity (MFI) was calculated using FlowJo V10 software.

**Construction of the *V. parahaemolyticus* genome dataset**. *V. parahaemolyticus* RefSeq genomes (assembly level "scaffold" or higher) were downloaded from NCBI on 24 June 2019. OrthoANI was performed according to Lee et al. 2016[67]. Briefly, each genome was fragmented into 1020 bp fragments. BLASTN was then performed against the *V. parahaemolyticus* RIMD 2210633 type strain, and OrthoANI values were calculated (Supplementary Data 1). Three genomes with OrthoANI values of <95% were removed from the dataset.

BLASTX was employed to identify the T6SS1 cluster proteins in *V. parahaemolyticus* genomes. Translated nucleotide sequences were aligned against the 24 T6SS1 cluster proteins of *V. parahaemolyticus* RIMD 2210633 (NP_797770.1 to NP_797793.1). A similarity percentage was calculated by dividing the bit-score value by two times the specific lengths of the cluster proteins (see below). Minimal similarity regarded as positive was defined as 50%. Bacterial genomes encoding at least 22 out of the 24 T6SS1 cluster proteins were regarded as harboring T6SS1 (T6SS1+). Bacterial genomes not encoding any of the T6SS1 cluster proteins were regarded as lacking T6SS1 (T6SS1−). A summary of the T6SS1 core proteins identified in *V. parahaemolyticus* genomes is provided in Supplementary Data 2.

**Comparative genomics analysis of *V. parahaemolyticus* BB22OP**. The proteins of *V. parahaemolyticus* BB22OP were aligned, using BLASTP, against the proteins

in the *V. parahaemolyticus* genome dataset. Similarity percentages were calculated by dividing the bit-score values by two times the lengths of the query proteins (see below). For each protein in *V. parahaemolyticus* BB22OP, the degree of similarity to the proteins in the *V. parahaemolyticus* genome dataset was evaluated, and the number of T6SS1[+] and T6SS1[−] genomes containing similar proteins (at least 50% similarity) was determined. Signal peptides and cleavage sites were predicted using SignalP 5.0[68].

Each *V. parahaemolyticus* BB22OP protein was evaluated for the presence of a potential immunity protein, either downstream or upstream, based on the following criteria: (a) the protein had an adjacent protein encoded within 50 bp on the same strand, (b) the adjacent protein was smaller than the protein, and (c) the length of the adjacent protein was at least 50 aa.

A candidate effector protein was identified based on the following criteria: (a) the length of the protein was at least 70 aa, (b) the protein had a potential immunity protein either downstream or upstream, (c) the protein was not similar to proteins in T6SS1[−] genomes, and (d) the protein did not contain a signal peptide.

**Calculation of the similarity percentage**. The bit-score calculated by the BLAST algorithm was used to approximate the overall similarity of the query protein to the subject protein. On average, each identical residue in the local alignment contributes ~2 bits to the overall bit score. Therefore, the similarity percentage was approximated by dividing the bit-score value by two times the length of the query protein. Notably, in some cases the approximated similarity percentage slightly deviates from the real value, as observed when query proteins are self-aligned.

**Identification of Tme-containing proteins**. The Position-Specific Scoring Matrix (PSSM) of the Tme domain was constructed using amino acids 161-310 of Tme1 from *V. parahaemolyticus* BB22OP (WP_015297525.1). Five iterations of PSI-BLAST[44] were performed against the reference protein database (a maximum of 500 hits with an expect value threshold of $10^{-6}$ were used in each iteration). A local database containing all RefSeq genomes from NCBI was constructed (last updated on 21 September 2019). The PSSM of the Tme domain was used to identify bacterial genomes that contained the Tme domain. Subsequently, the protein sequences and feature tables of Tme-containing genomes were retrieved from the local database.

RPS-BLAST[44] was used to identify Tme-containing proteins (Supplementary Data 5). The results were filtered using an expect value threshold of $10^{-8}$ and a minimal alignment length of at least 80 aa. Unique protein accessions located at the ends of genomic accessions were removed from subsequent analyses. To avoid duplications, unique protein accessions appearing in the same genome in more than one genomic accession were removed if the same downstream protein existed at the same distance.

The proteins encoded upstream of the Tme-encoding genes on the same strand were defined as upstream proteins, whereas proteins encoded downstream on the same strand were defined as downstream proteins. Protein sequences were analyzed to identify conserved domains (see below). Transmembrane topology and signal peptides were predicted using Phobius[52]. Signal peptides and cleavage sites were predicted using SignalP 5.0[68].

**Identification of Tme-containing genomes encoding T6SS**. RPS-BLAST[44] was employed to identify the T6SS core components in the Tme-containing bacterial genomes, as described before[26]. Briefly, the proteins were aligned against 11 COGs that were shown to specifically predict T6SS. COG3501 (VgrG) and COG0542 (ClpV) were not included as part of the T6SS core proteins because they were previously shown not to be specific to T6SS[1]. Bacterial genomes encoding at least nine out of the eleven T6SS core components were regarded as harboring T6SS. In addition, the number of genomes containing either Hcp or VgrG was evaluated (Supplementary Data 6).

**Construction of the phylogenetic tree of Tme domains**. Tme domain sequences were retrieved from the Tme-containing proteins and aligned using MUSCLE[69]. The evolutionary history was inferred using the Neighbor-Joining method[70]. The analysis involved 1,999 amino acid sequences and 98 conserved positions (95% site coverage). Evolutionary analyses were conducted in MEGA7[71].

**Illustration of conserved residues in the Tme domain**. Tme domain sequences were aligned using MUSCLE[69]. Aligned columns not found in *V. parahaemolyticus* BB22OP Tme1 (WP_015297525.1) were discarded. The Tme domain conserved residues were illustrated using the WebLogo 3 server[72] (http://weblogo.threeplusone.com).

**Identification of conserved domains and additional domains**. The Conserved Domain Database (CDD) and related information were downloaded from NCBI on June 29, 2019[73]. RPS-BLAST was employed to identify conserved domains and the output was processed using the Post-RPS-BLAST Processing Utility v0.1. The expect value threshold was set to $10^{-5}$. In addition, RPS-BLAST was used to identify the MIX[17,23] and FIX[26] domains.

**Clustering of downstream proteins using CLANS**. Downstream proteins located within 50 bp from Tme proteins were clustered using CLANS[47]. Information on DUF1240-containing proteins was retrieved from conserved domain analysis.

**Reporting summary**. Further information on research design is available in the Nature Research Reporting Summary linked to this article.

## Data availability
The experimental and computational data that support the findings of this research are available in this article and its supplementary information files, or upon request from the corresponding authors. The source data underlying Figs. 2, 3b–d, 5b–d, 6a–f, and Supplementary Figs. 2a–c, 3, 4a–d, 6, 7a–c, 8a–b, are provided as a Source Data file.

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

## Acknowledgements

This project received funding from the European Research Council (ERC) under the European Union's Horizon 2020 research and innovation program (Grant agreement No. 714224) to D.S., the Israel Science Foundation (ISF; grant no. 920/17) to D.S., the Israel Science Foundation (ISF; grant no. 818/18) to M.G., and the Margot Stoltz Foundation through the Faculty of Medicine grants of Tel Aviv University to M.G. We thank Michal Raz Karni and Rozanna Shevchencko for their excellent technical assistance, Udi Qimron for critical reading of the manuscript, and members of the Salomon and Bosis laboratories for fruitful discussions.

## Author contributions

E.B. and D.S. designed the study. C.M.F., K.K., and M.G. performed the experiments. E.B. performed the computational analyses. C.M.F, M.G, E.B., and D.S. analyzed the data. D.S. wrote the paper, and the other authors contributed to the writing. All authors read and approved the final version of the paper.

## Competing interests

The authors declare no competing interests.
