## [Peer Review File · Nature Communications]

Reviewers' comments:

Reviewer #1 (Remarks to the Author):

In this report, Fridman and colleagues report a methodology for type six effector identification and implement this approach to identify the Tme effector and its Tmi immunity protein. Numerous effector discovery approaches have been reported over the past decade and in the opinion of this reviewer, using comparative genomics followed by the application of additional criteria to filter results comes across as an extension/modification of the 2012 study by Russell et al (Cell Host & Microbe) rather than an entirely novel approach for effector discovery. Of the 17 potential hits identified in this study, only 1 conferred a fitness advantage to attacking cells versus susceptible prey using the 'surrogate system' suggesting the effector discovery approach presented herein may identify many false positives. I also find the novelty of the Tme effector itself to be underwhelming. As stated several times by the authors, Tme compromises cellular membranes in a similar manner to the VasX effector, which was characterized many years ago. The lack of biochemistry in this study makes it unclear what types of pores might be formed by Tme inserted into membranes and more in-depth characterization of membrane targeted effectors has been published previously (Lacourse et al., Nat Micro, 2018 and Mariano et al., Nat Commun, 2019). In summary, while I do find the data presented in this paper interesting, it does not represent a major advance in the field.

Reviewer #2 (Remarks to the Author):

Fridman et al. have developed a new approach to uncover T6SS effectors that overcomes some of the limitations of previous approaches. Although the approach comes with its own limitations, it is still complementary to the previous ones and allows to advance in one of the most difficult bottleneck problems of the field, the identification of novel T6SS effectors. The identification of novel antimicrobial effectors is always of interest for the scientific community for the possibility that opens for novel therapeutic targets to combat AMR.

The methodology is composed of two main steps, first being a genomic comparative analysis. In this case, they are 175 genomes of different isolates of the strain of interest publically available. This could not be the case for all strains, becoming an important limitation in other studies. Moreover, the analysis requires a bioinformatician to carry out comparative studies among the 175 genomes.

Secondly, the approach needs what they called a surrogate platform to screen the putative effectors selected in the previous analysis. This platform is a member of the species of interest that must contain an identical T6SS from the others selected in the study and which activation has to be known. This strain should also be able to be genetically modified to construct a T6SS positive strain (constitutively active) and a T6SS mutant (deletion mutant). Again, all these requisites for the surrogate platform could represent different limitations to the approach in some other systems.

However, in those cases where the comparative genome analysis is possible and the surrogate platform is available, the approach seems to work; although the number of newly effectors uncovered is quite limited. With a sample of 175 genomes available, only one effector was discovered although it results to be a new family of effectors and thus increase its importance.

I have some comments to be addressed and some suggestions:

- The authors could consider adding "targeting bacterial membranes" to the title
- Figure S1: a recent paper (Schneider 2019 – "Diverse roles of TssA-like proteins in the assembly of bacterial type VI secretion systems") has described the second tssA found in *Vibrio* clusters (in this figure tssA1b) as the gene encoding TagA with a N-term ImpA domain and a hydrophobic domain, thus tssA1b should be renamed as tagA1. tssA1a encodes a classical TssA with a N-term ImpA and a C-term VasJ and should be named as tssA1.
- Lane 104: In supplementary Dataset S2, the authors should explain how similarity is calculated and thus how numbers higher than 100% are possible.
- Lane 99: The authors stated that they are 175 genomes publically available but the excel file containing this information (supplementary dataset S1) contains 178. Please, correct this incongruency
- Lane 104: 2 genomes were disregarded from the analysis because they only have partial T6SS clusters. However, the subsequent study shows that only in the 77% of the genomes where Tme is present there are 9 or more core T6SS genes. That indicates that a high proportion of genomes with partial T6SS clusters contain effector-immunity pairs and thus these 2 genomes could be included in the study in the group of T6SS positives.
- In figure 1B, it is not clear which BB22OP proteins are represented. Is every row a different protein? If so, which one? Same for T6SS cluster. Does T6SS refer to T6SS1? Which proteins are represented in the last row named putative effector?
- Figure 1C, the right panel should be "no EI pair" instead of "not EI pair"
- Lane 143: Supplementary Dataset S2 should be changed to S1
- Lane 144: the authors should consider moving "candidate" afterwards "pairs" and change to "candidates".
- Supplementary Figure S2BC: It is surprising that prey cells can grow 2 log when they are not being killed by T6SS, but it seems to be the case for all the *Vibrio* competitions in the paper. It is not the case when the competition is against *E. coli*, could the authors comment in this fact?
- Supplementary Figure S2B: In the figure legend, the prey strain is dtdhAS/dvp1415-6 while in the figure is dvp1415-6. Could the authors comment on this?
- Supplementary Figure S2C: there is killing with the attacker strain harbouring the empty plasmid and no killing when the attacker strain is expressing effector V12G01_02265/0. This is not what it is expected and in fact, is not in accordance with the statement of the authors in lane 154. I am guessing these data have been swapped?
- Supplementary Figure S2C: In the case of effector 14465 from strain B5C30, the vgrG is cloned together with the effector and immunity gene. Is the whole vgrG included? Are these three genes one after each other in strain B5C30? How have they been cloned?
- Figure 2 shows clearly the toxic effect of the putative effector 15030 in the surrogate platform; it is very clear that an effector that shows toxicity in this assay is a good candidate. However, the authors should consider that this system might give false negatives and it is still possible that some of the other putative effectors are in fact T6SS effectors that fail to be delivered by the surrogate platform. A discussion about this point should be included. The authors should also consider adding information about the genomic context of these 7 genes encoding putative effectors. For the effector showing toxicity in this assay, it seems to be an orphan, but for the other 7 genes, nothing is said and being genetically linked to T6SS genes might indicate a false negative in this assay.
- Once Tme family has been characterised and conserved domains defined, the authors should specify the characteristic of Tme1, thus, the C-terminal domain (I am guessing is one of the unknown but independently of this, the formation should be included in the manuscript) and the genomic context (based on fig 3A, it will be in the 30% of others, but still, that should be included and easy for the reader to be found). Once Tme2 is introduced, the same information for Tme2 should be included.
- According to Supplementary dataset S4, there are 2 Tme proteins in BB22OP. The first with gene locus 15030 and product accession WP_015297525.1 (310 amino acids) that was identified in this study; and a second one WP_015296823.1 (284 amino acids) as a result of looking for Tme homologues in *V. parahaemolyticus* genomes. Is that correct? Are those proteins homologous? Why the gene encoding the second protein was not fished in the screening? Are immunity proteins

also homologous? Have the authors tested this second effector? In summary, why this is not discussed?

- Supplementary Figure S4: There is a delay in growth for all dhns strains, is that something that could be expected?
- Supplementary Dataset S5: The name of the T6SS protein containing the COGs used in this table should be included for clarification. The authors have not included ClpV for this study, could they comment on that? The number of T6SS core components is 13, is there any reason the authors only use 11, although they then use VgrG in a different column, this is confusing and should be better clarified
- Figure 4C: I wonder if the domain in Figure 4C are at scale, especially because VgrG is a big protein and in this figure is represented smaller than a PAAR domain that is normally quite small.
- Lane 246: Unknown C-terminal domains are the 97% of the cases, that should be stated in the text seems "many" doesn't represent this high percentage.
- Figure 4C: COG3515 is an ImpA domain, ImpA domain is present in TssA and TagA proteins, both structural components that have not yet been identified having a C-term effector domain. The frequency of this is very low 2/13206. What do the authors think about this? There is 0.95% and 0.01% of cases where the upstream gene is tssA or tagA respectively. Could it be that in these 2 cases the genes were wrongly annotated and they are not, in fact, part of the same protein?
- Lane 248: COG3515 should be also indicated as ImpA and the above case discussed.
- Figure 4D: It is very interesting that in 66% of the cases the upstream gene is hcp. Do the authors think that this could be an Hcp-dependent effector? Does it have the size for that? It is also found that 369/13206 have a C-term PAAR domain, in this case, it would be PAAR-dependent. Have the authors look for a relationship between the gene upstream and the C-term domain? If so, anything interesting to discuss? Do the orphan effectors (30.38% Others) have C-term related T6SS domains?
- Figure 5B: There is a second higher band appearing in the secretion of Tme2. Could the authors discuss this? Also, there is much less Tme2 in the T6SS negative strain and thus is difficult to assess the absence of Tme2 secretion. It could be the case that Tme2 is chaperoned by hcp1 and thus why is decreased in the mutant. Using a different T6SS structural mutant could help to solve this issue
- Effectors Tme1 and Tme2 work in the periplasm as clearly demonstrated in Fig 6A and B but it is not clear if the immunity proteins in this experiment have signal peptides (it should be specified and discussed). In the same way, it is not discussed at all the presence or absence of a signal peptide in the immunity proteins that belong to family Tmi. Checking in Supp. Dataset S4 and it seems that it is present only in some of them but not all, that should be further discussed.
- Lane 372, reference 50 is missing for VasX

Reviewer #3 (Remarks to the Author):

The manuscript by Fridman et al. reports on a new comparative genomics approach to identify novel T6SS E/I pairs. Furthermore, it characterizes one novel E/I class that was identified by employing this novel approach. They convincingly demonstrate that effectors of this new class likely act in the cell periplasm to disrupt the membrane, and is combated by expression of the cognate immunity protein. The mechanism of Tme action and Tmi immunity remain unclear. However, the work already represents a major advance without addressing this. This study is remarkably complete, the experiments are cleverly designed and well controlled, the data is clearly presented, and the manuscript is very well-written. Also the scope of the work is timely, represents a major advance, and should be of broad interest to the field. It is exceedingly rare that I receive manuscripts for review where I have no critical comments or experiments to suggest, and this is one of those manuscripts. I only have minor comments for the authors to consider:

1. For western blots of the C-terminally myc-tagged E/I proteins, which protein was tagged? The effector or the immunity protein? Were the C-terminally tagged proteins tested for functionality? I assume the expression constructs used in Figure 2 are untagged. Since protein degradation and turnover can be defined by the C-terminus of proteins, is it possible some of the E/I pairs tested for expression in Figure S3 are artificially high compared to the natively expressed proteins which are used in Figure 2?

2. Highlight the DxxK and D that flanks the TM helix in Figure 4A in some way (arrows or underline).

3. Line 243 – it would be valuable to provide some context for the statement that of the genomes that had Tme homologs “99.7% of the genomes harbored at least one hcp gene and one VgrG gene”. Of all genomes in the database, what percentage have one hcp gene and one VgrG gene?

4. It is difficult to see which strains are grouped together in the growth curves of Fig. S4. Do all of the hns mutants grow faster than the wildtype? Is this a well-documented phenomenon? It would be worth commenting on this phenotype in the figure legend or in the text.

5. For Fig. S7C – it looks like the hcp1 mutant grows a bit faster than the wildtype. Is this a reproducible result? If so, is it clear why? It would be worth commenting on this in the Figure legend or in the text.

6. Line 284 – substitute “wrongly” for “incorrectly”

7. Line 298 – substitute “visualize” for “observe”

8. Line 448 – substitute “available for genetic manipulations” for “amenable to genetic manipulation”

Response to reviewers' comments:

Reviewer #1 (Remarks to the Author):

In this report, Fridman and colleagues report a methodology for type six effector identification and implement this approach to identify the Tme effector and its Tmi immunity protein. Numerous effector discovery approaches have been reported over the past decade and in the opinion of this reviewer, using comparative genomics followed by the application of additional criteria to filter results comes across as an extension/modification of the 2012 study by Russell et al (Cell Host & Microbe) rather than an entirely novel approach for effector discovery.

The study by Russel et al (2012) was based on comparative proteomics, while our work is based on a comparative genomic analysis, followed by a functional surrogate screening platform. One main advantage of our approach is that it can be easily scaled up to analyze hundreds of genomes, which is less feasible using comparative proteomics.

Of the 17 potential hits identified in this study, only 1 conferred a fitness advantage to attacking cells versus susceptible prey using the 'surrogate system' suggesting the effector discovery approach presented herein may identify many false positives.

As explained throughout the text (lines 19-20, 81-83, 431-444), our pipeline comprised two stages: the first was a computational approach and the second was the surrogate T6SS platform. Therefore, a false-positive result should refer to a protein that passed both stages but later ruled out as a T6SS1-effector. Of the 4,724 BB22OP proteins that were analyzed, only one passed both stages. This candidate was later confirmed to be a genuine T6SS1 effector. Therefore, no false-positives were identified using this approach.

I also find the novelty of the Tme effector itself to be underwhelming. As stated several times by the authors, Tme compromises cellular membranes in a similar manner to the VasX effector, which was characterized many years ago. The lack of biochemistry in this study makes it unclear what types of pores might be formed by Tme inserted into membranes and more in-depth characterization of membrane targeted effectors has been published previously (Lacourse et al., Nat Micro, 2018 and Mariano et al., Nat Commun, 2019). In summary, while I do find the data presented in this paper interesting, it does not represent a major advance in the field.

Reviewer #2 (Remarks to the Author):

Fridman et al. have developed a new approach to uncover T6SS effectors that overcomes some of the limitations of previous approaches. Although the approach comes with its own limitations, it is still complementary to the previous ones and allows to advance in one of the most difficult bottleneck problems of the field, the identification of novel T6SS effectors. The identification of novel antimicrobial effectors is always of interest for the scientific community for the possibility that opens for novel therapeutic targets to combat AMR.

The methodology is composed of two main steps, first being a genomic comparative analysis. In this case, they are 175 genomes of different isolates of the strain of interest publically available. This could not be the case for all strains, becoming an important limitation in other studies. Moreover, the analysis requires a bioinformatician to carry out comparative studies among the 175 genomes.

Indeed, the approach presented in this study is not valid for all strains, however there are several bacterial strains for which sufficient number of high-quality genomic sequences are publically available. Furthermore, with the rapid increase in the number of available genomic sequences, we expect that in the upcoming years our approach will be valid for additional bacterial strains.

Secondly, the approach needs what they called a surrogate platform to screen the putative effectors selected in the previous analysis. This platform is a member of the species of interest that must contain an identical T6SS from the others selected in the study and which activation has to be known. This strain should also be able to be genetically modified to construct a T6SS positive strain (constitutively active) and a T6SS mutant (deletion mutant). Again, all these requisites for the surrogate platform could represent different limitations to the approach in some other systems.

Indeed, our approach relies on the ability to activate the T6SS1 in at least one member of the species of interest, and to be able to construct positive and negative T6SS1 strains by knocking-out relevant genes. Yet, this limitation is not specific to our approach, and any interrogation of the T6SS relies on these requirements. An important feature of our approach is that it allows studying T6SS effector repertoires in many strains by establishing a single member as a surrogate platform. By circumventing the necessity to genetically manipulate and activate the T6SS in each isolate, we provide the ability to characterize T6SS effector repertoires in multiple genomes.

However, in those cases where the comparative genome analysis is possible and the surrogate platform is available, the approach seems to work; although the number of newly effectors uncovered is quite limited. With a sample of 175 genomes available, only one effector was discovered although it results to be a new family of effectors and thus increase its importance.

In this work, we analyzed only a single *V. parahaemolyticus* genome (BB22OP). It was analyzed against 175 genomes that served as the Reference dataset. Analysis of this single genome revealed one effector, which led to the identification of a widespread family of T6SS effectors. As noted in the final section of the Discussion (lines 482-484), screening additional genomes against the same Reference dataset will likely result in the identification of additional novel effectors.

I have some comments to be addressed and some suggestions:

- The authors could consider adding “targeting bacterial membranes” to the title

We added “membrane-disrupting” to the title.

- Figure S1: a recent paper (Schneider 2019 – “Diverse roles of TssA-like proteins in the assembly of bacterial type VI secretion systems”) has described the second tssA found in *Vibrio* clusters (in this figure tssA1b) as the gene encoding TagA with a N-term ImpA domain and a hydrophobic domain, thus tssA1b should be renamed as tagA1. tssA1a encodes a classical TssA with a N-term ImpA and a C-term VasJ and should be named as tssA1.

We corrected Supplementary Figure 1 accordingly.

- Lane 104: In supplementary Dataset S2, the authors should explain how similarity is calculated and thus how numbers higher than 100% are possible.

We added an explanation of the calculation of the similarity percentage to the Methods section (lines 736-742). We note that in some cases the approximated similarity percentage may slightly deviate from the real value, as observed when query proteins are self-aligned.

- Lane 99: The authors stated that they are 175 genomes publically available but the excel file containing this information (supplementary dataset S1) contains 178. Please, correct this incongruency

We modified the text to clarify that 3 genomes with OrthoANI values of less than 95% were removed from the dataset (lines 707-708). We also added in Supplementary dataset 1 an appropriate remark next to those strains that were removed.

- Lane 104: 2 genomes were disregarded from the analysis because they only have partial T6SS clusters. However, the subsequent study shows that only in the 77% of the genomes

where Tme is present there are 9 or more core T6SS genes. That indicates that a high proportion of genomes with partial T6SS clusters contain effector-immunity pairs and thus these 2 genomes could be included in the study in the group of T6SS positives.

The two genomes that contained partial T6SS1 clusters were removed from the reference dataset because we were unable to determine with certainty whether these genomes represent T6SS1-positive or T6SS1-negative strains. Therefore, to be conservative, we decided to exclude them from subsequent analysis.

- In figure 1B, it is not clear which BB22OP proteins are represented. Is every row a different protein? If so, which one? Same for T6SS cluster. Does T6SS refer to T6SS1? Which proteins are represented in the last row named putative effector?

Indeed, each row in Fig. 1B represents a different protein. Shown are examples of the entire genome illustrating the various possibilities (Vp core genome proteins, T6SS1 cluster proteins, and putative effectors). T6SS referred to T6SS1. Figure 1b, including the legend, was modified to clarify these issues.

- Figure 1C, the right panel should be “no EI pair” instead of “not EI pair”

We corrected Figure 1c: “not-E/I” to “non-E/I”, short for non-effector/immunity pair.

- Lane 143: Supplementary Dataset S2 should be changed to S1

Supplementary Dataset 1 contains the OrthoANI analysis of *V. parahaemolyticus* genomes, while Supplementary Dataset 2 contains the summary of T6SS core proteins identified in *V. parahaemolyticus* genomes.

- Lane 144: the authors should consider moving “candidate” afterwards “pairs” and change to “candidates”.

Corrected.

- Supplementary Figure S2BC: It is surprising that prey cells can grow 2 log when they are not being killed by T6SS, but it seems to be the case for all the *Vibrio* competitions in the paper. It is not the case when the competition is against *E. coli*, could the authors comment in this fact?

Competition assay is performed on MLB media containing high salt concentration (3%), a medium in which E. coli are not growing well, especially at 30°C.

- Supplementary Figure S2B: In the figure legend, the prey strain is dtdhAS/dvp1415-6 while in the figure is dvp1415-6. Could the authors comment on this?

We corrected the Figure.

- Supplementary Figure S2C: there is killing with the attacker strain harbouring the empty plasmid and no killing when the attacker strain is expressing effector V12G01_02265/0. This is not what it is expected and in fact, is not in accordance with the statement of the authors in lane 154. I am guessing these data have been swapped?

The order of the data in the figure is correct. We amended the angle of the text in the Figure to avoid future confusion.

- Supplementary Figure S2C: In the case of effector 14465 from strain B5C30, the vgrG is cloned together with the effector and immunity gene. Is the whole vgrG included? Are these three genes one after each other in strain B5C30? How have they been cloned?

The whole vgrG was included, and the genes were cloned together in one operon from the beginning of vgrG to the end of the immunity. See text for details (lines 514-517). We also corrected p1b-Vp14465/0 to p1b+Vp14465/0 in the legend.

- Figure 2 shows clearly the toxic effect of the putative effector 15030 in the surrogate platform; it is very clear that an effector that shows toxicity in this assay is a good candidate. However, the authors should consider that this system might give false negatives and it is still possible that some of the other putative effectors are in fact T6SS effectors that fail to be delivered by the surrogate platform. A discussion about this point should be included.

We added a note to address this point in the text (lines 182-183): “Notably, any of the other effector candidates could be a true T6SS1 effector that failed to be utilized by the surrogate system.”

The authors should also consider adding information about the genomic context of these 7 genes encoding putative effectors. For the effector showing toxicity in this assay, it seems to be an orphan, but for the other 7 genes, nothing is said and being genetically linked to T6SS genes might indicate a false negative in this assay.

We added a note (lines 125-126) highlighting that “none of the genes encoding these proteins were found in proximity to T6SS-related genes (Supplementary Data 3)”.

- Once Tme family has been characterised and conserved domains defined, the authors should specify the characteristic of Tme1, thus, the C-terminal domain (I am guessing is one of the unknown but independently of this, the formation should be included in the manuscript) and the genomic context (based on fig 3A, it will be in the 30% of others, but still, that should be included and easy for the reader to be found). Once Tme2 is introduced, the same information for Tme2 should be included.

Indeed, the genomic context of both Tme1 and Tme2 is unrelated to T6SS, and their N-terminal domains are unknown. To clarify these points, the following amendments to the text were made: (1) in legends of Figs. 3A and 5A, “White arrows denote genes unrelated to T6SS.”; (2) in lines 232-233, “Analysis of Tme1 did not reveal any similarity to any known toxin domain or T6SS-associated domain”.; (3) in lines 288-289, “Tme2 maintained all of the abovementioned characteristics of a T6SS1 candidate effector, and did not contain identifiable domains other than Tme.”

- According to Supplementary dataset S4, there are 2 Tme proteins in BB22OP. The first with gene locus 15030 and product accession WP_015297525.1 (310 amino acids) that was identified in this study; and a second one WP_015296823.1 (284 amino acids) as a result of looking for Tme homologues in *V. parahaemolyticus* genomes. Is that correct? Are those proteins homologous? Why the gene encoding the second protein was not fished in the screening? Are immunity proteins also homologous? Have the authors tested this second effector? In summary, why this is not discussed?

WP_015296823.1 is a distant homolog of Tme1 and Tme2. It shares 8.8% similarity with Tme1 and 10.4% similarity with Tme2. WP_023623559.1 is the immunity protein of WP_015296823.1 and it shows no similarity to Tmi1 and very low similarity to Tmi2 (4.9% with E-value of 0.31). This protein was not identified in our comparative genomics analysis because it has identical copies in two T6SS1-minus genomes that are part of our Reference dataset (D3112 and FORC_071). It implies that WP_015296823.1 is not secreted by T6SS1, and our unpublished results suggest that it is secreted by T6SS2. Although this result supports our argument regarding the specificity of our T6SS1 screen and that Tme proteins are T6SS effectors, we decided not to include this result here because this work focuses on T6SS1. We intend to publish this result in the future as part of an analysis of T6SS2 effectors.

- Supplementary Figure S4: There is a delay in growth for all dhns strains, is that something that could be expected?

The Δ hns strains are growing faster than hns+ strains in Supplementary figure 4. To clarify this point, we denote each cluster of growth curves in Supplementary Figure 4 with Δ hns or hns+. We note that hns is a global regulator and in different strains it may affect bacterial growth and behavior differently. Therefore, it is difficult to anticipate what effect its deletion will have on growth. Nevertheless, the results indicate that there no difference in growth within strains with the same hns background. See text: “deletions of neither hcp1 nor hcp2 affected bacterial growth in BB22OP wild-type or Δ hns backgrounds” (lines 209-211), and “deletion of tme1 did not affect BB22OP growth” (line 224).

- Supplementary Dataset S5: The name of the T6SS protein containing the COGs used in this table should be included for clarification. The authors have not included ClpV for this study, could they comment on that? The number of T6SS core components is 13, is there any reason the authors only use 11, although they then use VgrG in a different column, this is confusing and should be better clarified

The names of T6SS core proteins were added to Supplementary Dataset 5. Our usage of 11 core components was based on the analysis of Boyer et al (2009) who showed that COG3501 (VgrG) and COG0542 (ClpV) are not specific to T6SS. We added a clarification (lines 768-770): “COG3501 (VgrG) and COG0542 (ClpV) were not included as part of the T6SS core proteins because they were previously shown not to be specific to T6SS.”

- Figure 4C: I wonder if the domain in Figure 4C are at scale, especially because VgrG is a big protein and in this figure is represented smaller than a PAAR domain that is normally quite small.

Domain sizes are not to scale. A clarification was added to the legend.

- Lane 246: Unknown C-terminal domains are the 97% of the cases, that should be stated in the text seems “many” doesn’t represent this high percentage.

The word “many” was replaced by “most” (line 246).

- Figure 4C: COG3515 is an ImpA domain, ImpA domain is present in TssA and TagA proteins, both structural components that have not yet been identified having a C-term effector domain. The frequency of this is very low 2/13206. What do the authors think about this? There is 0.95% and 0.01% of cases where the upstream gene is tssA or tagA respectively. Could it be that in these 2 cases the genes were wrongly annotated and they are not, in fact, part of the same protein?

We also noticed this interesting result. This is probably not a mistake as we found this fusion protein of ImpA and Tme in several genomes (many of these genomes are not in RefSeq, which we used in the current study). We modified COG3515 to ImpA in Figure 4c, and we added a short paragraph in the Discussion relating to this result (lines 400-410).

- Lane 248: COG3515 should be also indicated as ImpA and the above case discussed.

Corrected (line 248).

- Figure 4D: It is very interesting that in 66% of the cases the upstream gene is hcp. Do the authors think that this could be an Hcp-dependent effector? Does it have the size for that? It is also found that 369/13206 have a C-term PAAR domain, in this case, it would be PAAR-dependent. Have the authors look for a relationship between the gene upstream and the C-term domain? If so, anything interesting to discuss?

Indeed, it is possible that when Tme are encoded directly downstream of Hcp they are dependent on it for T6SS-mediated delivery. However, as the reviewer noted, in other cases Tme is fused to PAAR, VgrG, MIX, or ImpA. Moreover, some Tme domains are part of large proteins (up to 948 amino acids) that are too large to fit in the Hcp tube. Therefore, it is not possible to reach a general conclusion regarding the preferable secretion mechanism of Tme domains, if one exists.

We did not perform an in-depth analysis to determine if there is a connection between the gene upstream of Tme, and the N-terminus (we assume that was the reviewer's intention since Tme itself is the C-terminal domain) of Tme-containing proteins.

Do the orphan effectors (30.38% Others) have C-term related T6SS domains?

We did not analyze the domain content of the orphan Tme effectors specifically. However, we note that only ~400 Tme have T6SS-related N-terminal domains (Fig. 4c), which is ~10% of the orphan occurrences (Fig. 4d). Therefore, even if such distribution exists, it would not allow us to comment on possible mechanisms of secretion for the orphan Tme.

- Figure 5B: There is a second higher band appearing in the secretion of Tme2. Could the authors discuss this?

We made a mistake in the ladder shown in the bottom panel (cells), which has been corrected (see Figure 5b). The expected size of the protein, as can now be seen in the "cells" panel, is slightly above the 45 kDa marker. Therefore, the upper band in the

secretion panel is actually the expected protein, whereas the lower band is possibly a degradation product. We apologize for the inconvenience and confusion caused by this mistake.

Also, there is much less Tme2 in the T6SS negative strain and thus is difficult to assess the absence of Tme2 secretion. It could be the case that Tme2 is chaperoned by hcp1 and thus why is decreased in the mutant. Using a different T6SS structural mutant could help to solve this issue

We agree with the reviewer that it is possible that Tme2 stability is affected by Hcp1. However, the secreted band detected in the T6SS1⁺ is quite clear whereas we detect no signal in the T6SS⁻ sample even when adjusting brightness/contrast of the image. Nevertheless, we believe that determining the mechanism of secretion of an individual Tme effector (which we have demonstrated is a bona fide T6SS1 effector also in competition assays) is beyond the scope of the current work.

- Effectors Tme1 and Tme2 work in the periplasm as clearly demonstrated in Fig 6A and B but it is not clear if the immunity proteins in this experiment have signal peptides (it should be specified and discussed). In the same way, it is not discussed at all the presence or absence of a signal peptide in the immunity proteins that belong to family Tmi. Checking in Supp. Dataset S4 and it seems that it is present only in some of them but not all, that should be further discussed.

While most Tmi proteins do not contain a predicted signal peptide, they do have N-terminal transmembrane helices that can serve as membrane anchors. We now note this in the Discussion (lines 414-416).

- Lane 372, reference 50 is missing for VasX

The reference was added.

Reviewer #3 (Remarks to the Author):

The manuscript by Fridman et al. reports on a new comparative genomics approach to identify novel T6SS E/I pairs. Furthermore, it characterizes one novel E/I class that was identified by employing this novel approach. They convincingly demonstrate that effectors of this new class likely act in the cell periplasm to disrupt the membrane, and is combated by expression of the cognate immunity protein. The mechanism of Tme action and Tmi immunity remain unclear. However, the work already represents a major advance without addressing this. This study is remarkably complete, the experiments are cleverly designed and well controlled, the data is clearly presented, and the manuscript is very well-written. Also the scope of the work is timely, represents a major advance, and should be of broad interest to the field. It is exceedingly rare that I receive manuscripts for review where I have no critical comments or experiments to suggest, and this is one of those manuscripts. I only have minor comments for the authors to consider:

1. For western blots of the C-terminally myc-tagged E/I proteins, which protein was tagged? The effector or the immunity protein?

As detailed in the Methods section (lines 512-514), the protein encoded by the downstream gene of each tested gene-pair was fused to a C-terminal Myc tag.

Were the C-terminally tagged proteins tested for functionality? I assume the expression constructs used in Figure 2 are untagged. Since protein degradation and turnover can be defined by the C-terminus of proteins, is it possible some of the E/I pairs tested for expression in Figure S3 are artificially high compared to the natively expressed proteins which are used in Figure 2?

The plasmids used in Supplementary Figure 3 are the same ones used in Figure 2. In these plasmid, as mentioned above, the downstream gene is fused to a C-terminal myc tag to allow detection of inducible expression for each cassette. To avoid confusion, Figure 2 and its legend now indicate the myc tag, as in Supplementary Figure 3.

2. Highlight the DxxK and D that flanks the TM helix in Figure 4A in some way (arrows or underline).

Cyan ovals were added above the conserved residues in Figure 4a.

3. Line 243 – it would be valuable to provide some context for the statement that of the

genomes that had Tme homologs “99.7% of the genomes harbored at least one hcp gene and one VgrG gene”. Of all genomes in the database, what percentage have one hcp gene and one VgrG gene?

46.1% of all available RefSeq genomes (>170,000) harbor at least one Hcp and at least one VgrG. These data are part of a large analysis of secretion systems that we are currently working on. We think that such an analysis is beyond the scope of the current study. However, if the reviewer or the editor feels that this information is essential, then we suggest to include it in the text as “our unpublished results”.

4. It is difficult to see which strains are grouped together in the growth curves of Fig. S4. Do all of the hns mutants grow faster than the wildtype? Is this a well-documented phenomenon? It would be worth commenting on this phenotype in the figure legend or in the text.

The Δ hns strains are growing faster than hns+ strains in Supplementary Figure 4. To clarify this point, we denote each cluster of growth curves in Supplementary Figure 4 with Δ hns or hns+. We note that hns is a global regulator and in different strains it may affect bacterial growth and behavior differently. Therefore, it is difficult to anticipate what effect its deletion will have on growth. Nevertheless, the results indicate that there no difference in growth within strains with the same hns background.

5. For Fig. S7C – it looks like the hcp1 mutant grows a bit faster than the wildtype. Is this a reproducible result? If so, is it clear why? It would be worth commenting on this in the Figure legend or in the text.

Yes, this is a reproducible result, yet we do not know the reason behind it. Nevertheless, since it grows faster than the WT, the lack of effector-mediated killing when using this mutant as an attacker in competition assays did not stem from slower growth compared to the WT attacker.

6. Line 284 – substitute “wrongly” for “incorrectly”

Amended.

7. Line 298 – substitute “visualize” for “observe”

Amended.

8. Line 448 – substitute “available for genetic manipulations” for “amenable to genetic

manipulation”

Amended.